# An IMiD-induced SALL4 degron system for selective degradation of target proteins

Satoshi Yamanaka [1], Yuki Shoya[1], Saya Matsuoka[1], Hisayo Nishida-Fukuda [2], Norio Shibata [3] & Tatsuya Sawasaki [1]✉

Regulating the amount of proteins in living cells is a powerful approach for understanding the functions of the proteins. Immunomodulatory drugs (IMiDs) induce the degradation of neosubstrates by interacting with celebron (CRBN) in the cullin E3 ubiquitin ligase complex (CRL4$^{CRBN}$). Here, we developed the IMiD-dependent Sal-like protein 4 (SALL4) degron (S4D) system for chemical protein knockdown. In transient assays, an N- or C-terminal S4D tag induced the degradation of proteins localized to various subcellular compartments, including the plasma membrane. The activity of luciferase-S4D was reduced by 90% within 3 h of IMiD treatment. IMiD treatment reduced the expression of endogenous S4D-fused RelA and IκBα in knock-in (KI) experiments. Interestingly, the IκBα knockdown suggested that there may be another, unknown mechanism for RelA translocation to the nucleus. Furthermore, 5-hydroxythalidomide as a thalidomide metabolite specifically degradated S4D-tagged protein. These results indicate that the S4D system is a useful tool for cellular biology.

[1] Division of Cell-Free Sciences, Proteo-Science Center, Ehime University, Matsuyama 790-8577, Japan. [2] Department of Genome Editing, Institute of Biomedical Science, Kansai Medical University, Hirakata 573-1010, Japan. [3] Department of Nanopharmaceutical Sciences, Nagoya Institute of Technology, Nagoya 466-8555, Japan. ✉email: sawasaki@ehime-u.ac.jp

Controlling the expression of target proteins is a powerful approach to understand the biological functions of proteins. Protein-knockdown technology based on small molecule compounds has many advantages, such as simplicity, the ability to be used in many cell types, reversibility, and speed[1]. Many systems, including the auxin-inducible degron (AID) system, combine genetic and chemical strategies to achieve protein knockdown[2–4]. These systems combine the specificity of genetic approaches with the ease of use of small molecule compounds[1]. However, they require modified proteins that are tagged with specific amino acid sequences. In addition, researchers need to consider the effects of the additional tag on the localization or activity of the protein of interest. These systems therefore offer some improvements for the utilization in cell biology. Further improvement of degron system is an important issue to enhance the convenience of the protein-knockdown technology in biological field.

Immunomodulatory drugs (IMiDs) such as thalidomide and its derivatives interact with cereblon (CRBN) in the cullin E3 ubiquitin ligase complex (CRL4$^{CRBN}$)[5] and change its substrate specificity[6–8]. IMiD-dependent substrates such as IKZF3 and Sal-like protein 4 (SALL4), which are degraded by IMiD treatment, are called "neosubstrates"[6,8]. IKZF3 and SALL4 interact with the CRBN-IMiD complex through a single C2H2 zinc finger domain (ZNF)[8–10]. These observations led us to hypothesize that IMiD-dependent degradation could be applied to a novel system for the regulation of protein expression that would not require the co-expression of an additional E3 ligase, unlike the AID system. Indeed, in 2019, a peptide degron tag using the second zinc finger domain (ZNF2) of IKZF3 was reported. The tagged proteins were degraded by IMiD treatment in cultured cells and in xenograft models of kidney, breast, and brain cancer[11]. However, its use was still limited to the C-terminus of the protein of interest and overexpressed tagged proteins, not endogenous proteins[11].

Nuclear factor (NF)-κB is a pivotal transcription factor in inflammatory and immune responses[12,13]. In resting cells, IκB binds to NF-κB, blocking its nuclear localization signal (NLS) and causing it to be retained within the cytoplasm. In response to inflammatory stimuli such as TNF-α, IκB kinase (IKK) becomes activated and phosphorylates IκB, leading to its proteasomal degradation[12,14]. In canonical NF-κB signal transduction, the main form of NF-κB is a heterodimer composed of p65/RelA and p50, while the main IκB isoform is IκBα[12,14]. The mechanisms of NF-κB activation are transiently regulated by many proteins, including kinases, ubiquitin E3 ligases, and deubiquitinating enzymes[12,14–16]. We hypothesized that a protein degradation system based on IMiD would be useful for analyzing the dynamic and complicated biological phenomenon of NF-κB signal transduction.

In previous reports, it has been reported that several IMiDs have specificity for several neosubstrates[7,17]. 5-Hydroxythalidomide (5-HT) known as a thalidomide metabolite induces protein degradation of SALL4 but does not IKZF1[18]. In addition, 5-HT more strongly degrades SALL4 than thalidomide[18]. This suggests that the utilization of 5-HT provides the construction of a highly specific and efficient degradation system without the extra degradation of IKZF1 and IKZF3.

Herein, we developed a Sal-like protein 4 (SALL4) degron (S4D) tag as an IMiD-induced protein degradation system, and showed that the S4D system can be available to analyze proteins expressing transiently at various subcellular compartments. In addition, we integrated the S4D tag into IκBα or RelA by genome editing. Endogenous tagged IκBα or RelA protein was efficiently degraded by IMiD treatment. Finally, we showed that the combination 5-HT and the S4D system is a more powerful tool for analyzing S4D-tagged proteins. Taken together, the S4D system

can be a useful system to analyze both exogenous and endogenous S4D-tagged proteins.

## Results

**Construction of the IMiD-induced SALL4 degron tag.** In previous studies, SALL4 has been shown to interact with CRBN-IMiD through its ZNF2 domain, resulting in its degradation[8,9]. As shown in Fig. 1a, b, we prepared three SALL4-ZNF2 fusion proteins, designated Venus-m1 (405–437), -m2 (410–437), and -m3 (410–432), and examined whether the additional sequence induced an interaction with CRBN and degradation of the tagged protein in the presence of IMiD. In in vitro binding assays using AlphaScreen and a wheat cell-free system (Supplementary Fig. 1a), Venus-m1 and Venus-m2 interacted with CRBN in the presence of IMiD, but Venus-m3 did not (Supplementary Fig. 1b, c). Next, we examined the protein degradation in HEK293T-CRBN$^{-/-}$ cells expressing FLAG-CRBN by using the AGIA-tag system, which is a highly sensitive tag based on a rabbit monoclonal antibody[19]. Consistent with the in vitro binding assay results, Venus-m1 and -m2 were decreased by IMiD treatment, but Venus-m3 was not (Fig. 1c). Because the m2 sequence is shorter than m1, m2 was selected as the SALL4 degron tag and named S4D. Furthermore, the inhibitor treatment with MG132 or MLN4924, a proteasome or Nedd8-activating enzyme inhibitor, respectively, completely rescued the degradation of Venus-S4D (Fig. 1d), showing that this downregulation depends on the 26 S proteasome and the cullin E3 ligase complex. In previous studies, it was shown that SALL4 with a mutation of Gly416 in the ZNF2 to Ala cannot interact with CRBN-IMiD and cannot be degraded by IMiD treatment[8,9]. Immunoblot analysis showed that Venus-m2-G416A was not degraded upon IMiD treatment (Fig. 1e). Next, we investigated whether IMiD-dependent protein degradation of AGIA-Venus-S4D is induced by also endogenous CRBN because exogenous FLAG-CRBN was increased by IMiD treatment. In parental HEK293T cells, endogenous CRBN induced protein degradation of AGIA-Venus-S4D in the presence of IMiD (Fig. 1f). These results suggest that S4D-tagged protein is degraded by IMiD treatment.

Next, IKZF3 degron (I3D) was cloned into the C-terminus of Venus for comparison of I3D and S4D (Supplementary Fig. 2a). S4D-tagged Venus was degraded similarly to I3D-tagged Venus (Supplementary Fig. 2b). It was reported that an N-terminal IKZF3 degron (I3D) tag did not induce degradation[11]. Therefore, to examine whether the position of the S4D tag affected the degradation of the tagged proteins, expression vectors encoding S4D-Venus or Venus-S4D were constructed, which S4D tag was fused to N- or C-terminus of Venus protein, respectively (Supplementary Fig. 2c). Immunoblot analysis showed that S4D-Venus was degraded in response to IMiD treatment, similar to Venus-S4D (Supplementary Fig. 2d), indicating that the S4D system can be used to induce the degradation of both N- and C-terminally tagged proteins.

To more quantitatively characterize the S4D, Firefly luciferase (FLuc)-AGIA and FLuc-S4D-AGIA vectors were constructed (Fig. 1g). Immunoblot analysis showed that IMiD treatment decreased the expression of FLuc-S4D-AGIA but not FLuc-AGIA (Fig. 1h). In addition, luciferase activity in cell lysates treated with IMiD was reduced in an S4D-dependent manner (Fig. 1i). We next characterized the dose- and time-dependency of the S4D-induced protein degradation. The strength of the degradation was pomalidomide > lenalidomide > thalidomide, and 1–10 μM of these IMiDs dramatically reduced the expression of FLuc-S4D-AGIA (Fig. 1j). The IMiD treatment showed effect on the luciferase activity within an hour and then decreased by 90% in 3 h (Fig. 1k). Furthermore, MLN4924 completely rescued the

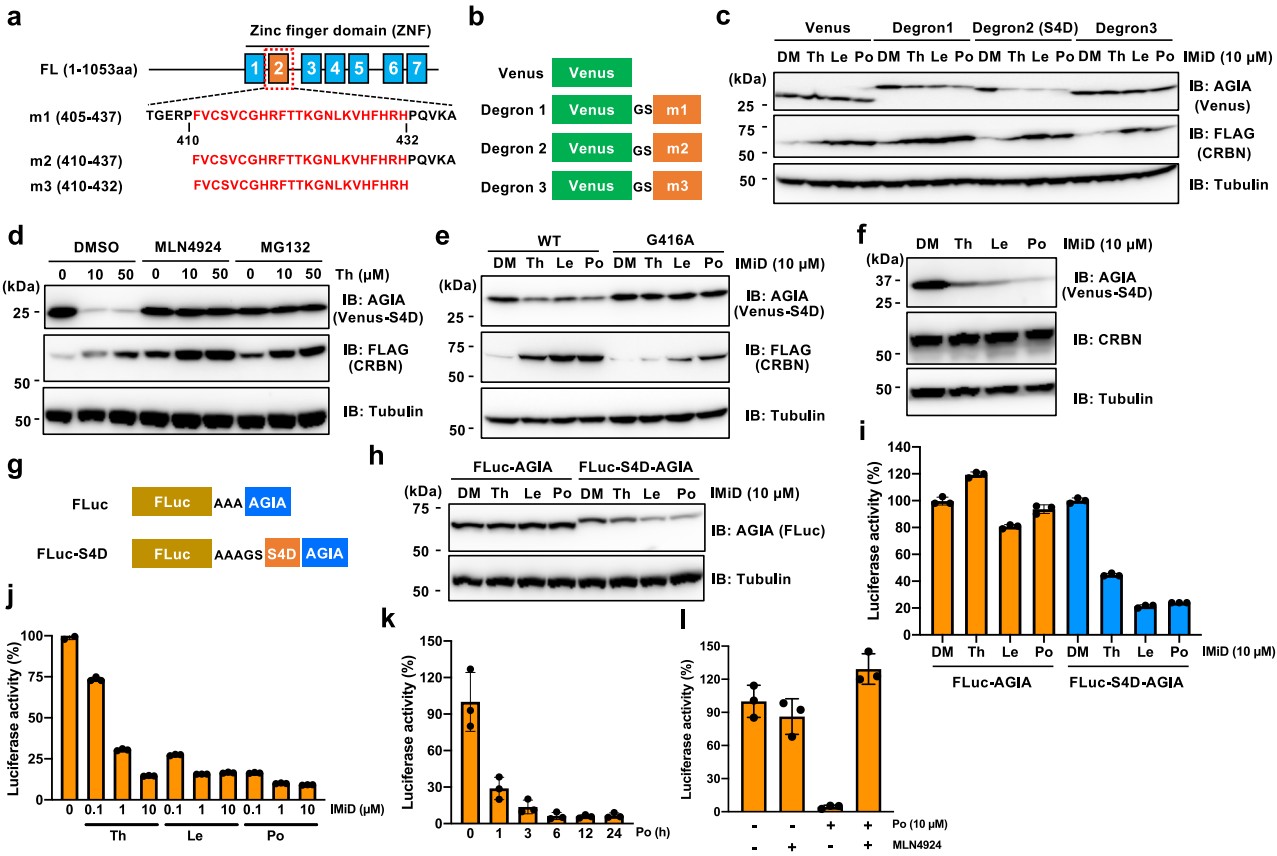

**Fig. 1 Construction of the IMiD-induced SALL4 degron (S4D) tag. a** Amino acid sequence of IMiD-induced SALL4 degron. **b** Schematic diagram of the ORFs of the expression vector encoding Venus-SALL4 degron. **c** Immunoblot analysis of Venus-SALL4 degron protein. CRBN$^{-/-}$ HEK293T cells expressing AGIA-Venus-m1, -m2 (S4D), or -m3 and FLAG-CRBN were treated with DMSO (DM), thalidomide (Th), lenalidomide (Le), or pomalidomide (Po) for 16 h. **d** Immunoblot analysis of AGIA-Venus-S4D protein in CRBN$^{-/-}$ HEK293T cells expressing AGIA-S4D-WT and FLAG-CRBN treated with DM or Th in the presence of DM, MG132, or MLN4924 for 9 h. **e** Immunoblot analysis of Venus-S4D-WT or -G416A. CRBN$^{-/-}$ HEK293T cells expressing AGIA-Venus-S4D-WT or -G416A and FLAG-CRBN were treated with DM, Th, Le, or Po for 16 h. **f** Immunoblot analysis of Venus-S4D in HEK293T cells. HEK293T cells expressing AGIA-Venus-S4D were treated with DM, Th, Le, or Po for 16 h. **g** Schematic diagram of the ORF in an expression vector encoding FLuc and FLuc-S4D. **h** Immunoblot analysis of FLuc and FLuc-S4D. HEK293T cells expressing FLuc-AGIA or FLuc-S4D-AGIA were treated with DM, Th, Le, or Po for 16 h. **i, j** FLuc luciferase activity in lysates of HEK293T cells expressing FLuc-AGIA or FLuc-S4D-AGIA treated with DM, Th, Le, or Po for 16 h. **k** FLuc luciferase activity in lysates of HEK293T cells expressing FLuc-S4D-AGIA and treated with DMSO or 10 μM pomalidomide for the indicated times. **l** FLuc luciferase activity in lysates of HEK293T cells expressing FLuc-S4D-AGIA and treated with DM or Po in the presence of DM or MLN4924 for 12 h. Error bars in **i-l** represent the mean ± SD ($n = 3$).

IMiD-induced degradation of FLuc-S4D-AGIA (Fig. 1l). These results demonstrate that IMiD treatment for more than 3 h resulted in the degradation of 90% of S4D-tagged protein, suggesting that the S4D system can dramatically reduce the expression of tagged proteins.

**S4D-tagged proteins are degraded in various subcellular compartments.** Next, we evaluated whether S4D can be used for knockdown of other proteins. It has been reported that proteins with various subcellular localizations, including the endoplasmic reticulum (ER) membrane[20,21], the mitochondrial outer membrane[22,23], and the plasma membrane[24,25], can be targeted for proteasomal degradation by 26S. Thus, the S4D system was applied to proteins with different subcellular localizations. As shown in Fig. 2a, p53 (nucleus)[26], stimulator of interferon genes (STING; ER membrane)[27], dopamine receptor 1 (DRD1; plasma membrane)[28], and mitochondrial antiviral signaling protein (MAVS; mitochondrial outer membrane)[29] were selected, and expression vectors were constructed in which the S4D tag was inserted into the C-terminus of each gene. After transfection of HEK293T-CRBN$^{-/-}$ cells with FLAG-CRBN and each construct,

immunoblot analysis showed that pomalidomide treatment dose-dependently decreased the expression of each protein analyzed (Supplementary Fig. 3). In addition, the localization and degradation of these proteins were evaluated in transiently transfected HeLa cells. All proteins localized to the expected compartments mainly (Fig. 2b), and degradation of each of the proteins upon IMiD treatment was observed by immunoblot analysis (Fig. 2c). MLN4924 rescued the reduction of protein degradation in HEK293T-CRBN$^{-/-}$ cells expressing FLAG-CRBN (Fig. 2d). Furthermore, pomalidomide-dependent protein degradation was observed in cells expressing endogenous CRBN (Fig. 2e). By contrast, in CRBN$^{-/-}$ cells, pomalidomide did not induce protein degradation (Fig. 2e), confirming that the reduction in protein expression was dependent on CRL4$^{CRBN}$ in the presence of pomalidomide. Then, to examine whether the S4D system can be also applied for degradation of protein localized at golgi apparatus, we constructed the expression vector encoding S4D-tagged Golgin subfamily A member 2 (GOLGA2/GM130) (Fig. 2f), which is known as a protein localized at cis-golgi network[30]. Immunofluorescence staining showed that GM130-S4D-AGIA was localized together with endogenous GM130 mainly (Fig. 2g)

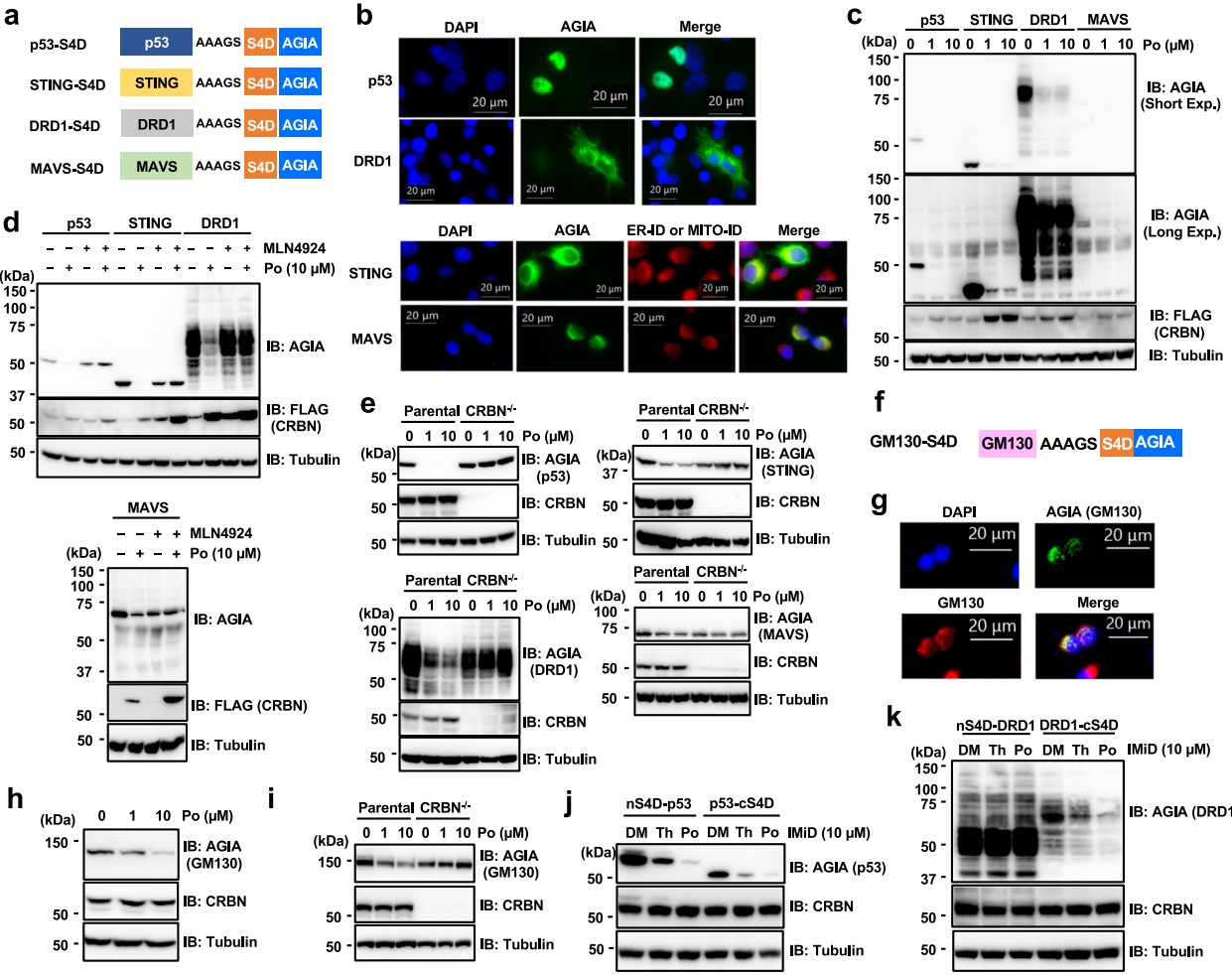

**Fig. 2 IMiD treatment induced the degradation of proteins with various subcellular localizations. a** Schematic diagram of the ORFs of the expression vectors encoding p53-S4D-AGIA, STING-S4D-AGIA, DRD1-S4D-AGIA, and MAVS-S4D-AGIA. **b** Localization of transiently expressed proteins. Immunofluorescent staining of p53-S4D-AGIA, STING-S4D-AGIA, DRD1-S4D-AGIA, or MAVS-S4D-AGIA in HeLa cells. Scale bars, 20 μm. **c** Immunoblot analysis of the dose-dependent degradation of proteins with various subcellular localizations. HeLa cells expressing p53-S4D-AGIA, STING-S4D-AGIA, DRD1-S4D-AGIA, or MAVS-S4D-AGIA and FLAG-CRBN were treated with DMSO (DM) or pomalidomide (Po) for 16 h. **d** Immunoblot analysis of the indicated proteins in various subcellular localizations. CRBN$^{-/-}$ HEK293T cells expressing p53-S4D-AGIA, STING-S4D-AGIA, DRD1-S4D-AGIA, or MAVS-S4D-AGIA and FLAG-CRBN were treated with DM or Po in the presence of DM or 2 μM MLN4924 for 9 h. **e** Immunoblot analysis of CRBN dependency on protein degradation of proteins with various subcellular localizations. Parental or CRBN$^{-/-}$ HEK293T cells expressing p53-S4D-AGIA, STING-S4D-AGIA, DRD1-S4D-AGIA, or MAVS-S4D-AGIA were treated with DM or Po for 16 h. **f** Schematic diagram of the ORFs of the expression vectors encoding GM130-S4D-AGIA. **g** Localization of transiently expressed GM130. Immunofluorescent staining of GM130-S4D-AGIA in HeLa cells. Scale bars, 20 μm. **h** HeLa cells expressing GM130-S4D-AGIA were treated with DMSO (DM) or pomalidomide (Po) for 16 h. **i** Immunoblot analysis of CRBN dependency on protein degradation of GM130-S4D-AGIA. GM130-S4D-AGIA degradation was analyzed by same procedure in Fig. 2e. **j, k** Immunoblot analysis of N- or C-terminally tagged p53 (**j**) or DRD1 (**k**). HEK293T cells expressing S4D-p53-AGIA, p53-S4D-AGIA, S4D-DRD1-AGIA, or DRD1-S4D-AGIA were treated with DM, thalidomide (Th), or Po for 16 h.

and protein degradation of GM130-S4D-AGIA was induced by IMiD treatment (Fig. 2h). In addition, CRBN dependency on the protein degradation of GM130-S4D-AGIA was confirmed by using HEK293T-CRBN$^{-/-}$ cells (Fig. 2i). Consistent with results shown in Supplementary Fig. 2d, although there was differential mobility of N- or C-terminal S4D-tagged protein by unknown reasons, IMiD treatment induced the degradation of both S4D-p53-AGIA and p53-S4D-AGIA in HEK293T cells (Fig. 2j). DRD1 is a G-protein-coupled receptor whose N-terminus is thought to be located on the outside surface of the plasma membrane. Interestingly, S4D-DRD1-AGIA was not degraded by IMiD treatment, although DRD1-S4D-AGIA was (Fig. 2k). Furthermore, in HeLa cells stably expressing DRD1-S4D-AGIA same form of Fig. 2a, DRD1-S4D-AGIA was localized at plasma membrane (Supplementary Fig. 4a) mainly and induced

degradated by IMiD treatment (Supplementary Fig. 4b). These results suggest that the S4D system can be used for proteins that localize to various subcellular compartments, including the plasma membrane but it is required for more detail analyses to demonstrate whether S4D-tagged protein degradations happen on each subcellular compartment.

**Transient transfection assays using the S4D system.** The adaptor protein STING recognizes cyclic GMP-AMP (cGAMP) generated by cGAS and activates type I interferon (IFN) signal transduction[31]. We first transiently expressed STING-AGIA or STING-S4D-AGIA in HEK293T cells, and performed luciferase reporter assays using an interferon-stimulated response element (ISRE). Consistent with previous reports[31], because endogenous STING is not expressed in HEK293T cells, cGAMP treatment did

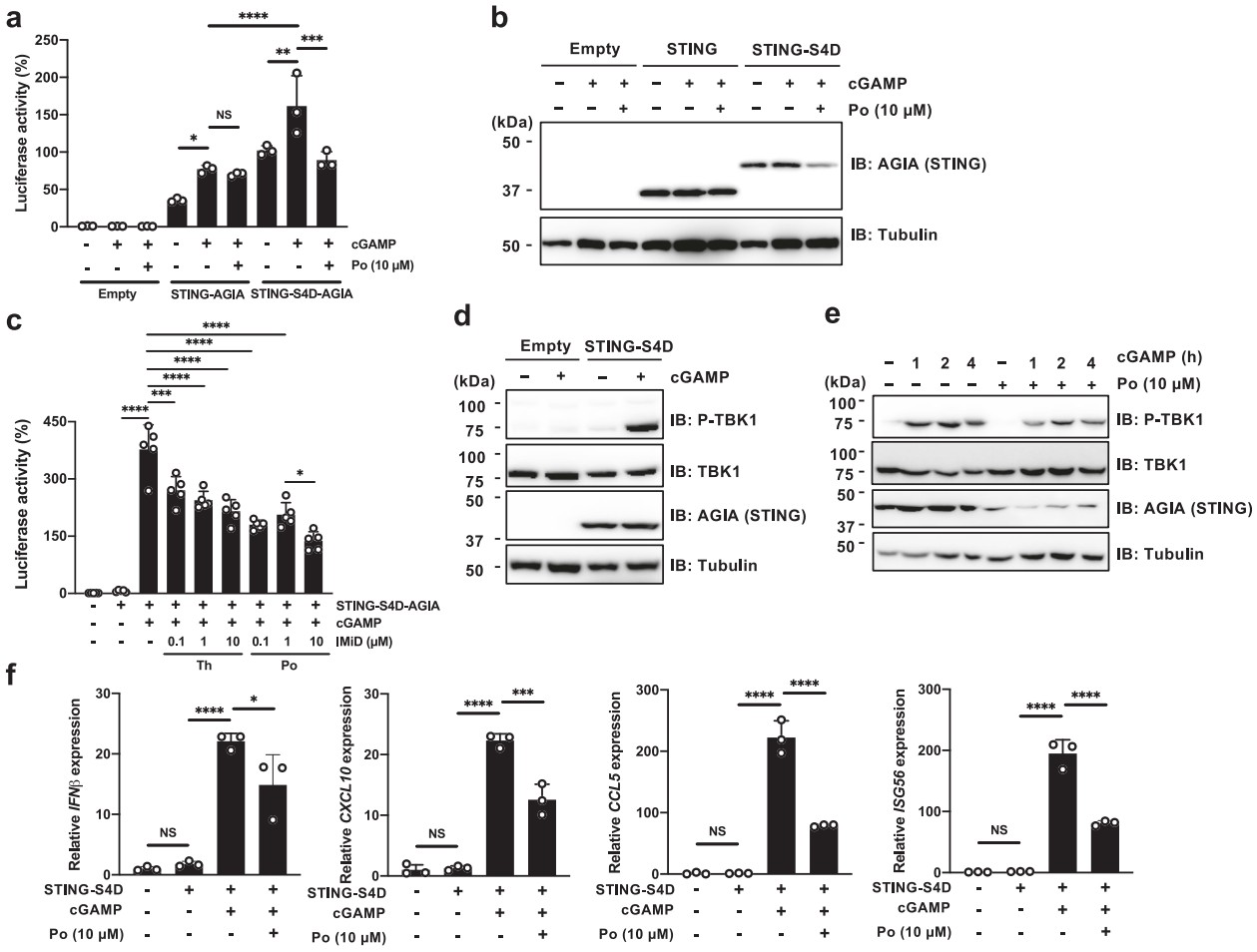

**Fig. 3 Analysis using the S4D system in transient transfection assays. a** Effect of pomalidomide-induced STING degradation on cGAMP-induced-ISRE luciferase reporter assay. HEK293T cells expressing STING-AGIA or STING-S4D-AGIA were treated with DMSO (DM) or pomalidomide (Po) for 16 h, and then stimulated with cGAMP for 3 h. **b** The expression of STING-AGIA or STING-S4D-AGIA in cells treated as described in Fig. 3a was analyzed by immunoblot. **c** Dose-dependent effects of cells treated as described in Fig. 3a. HEK293T cells expressing STING-S4D-AGIA were treated with DM, thalidomide (Th), or pomalidomide (Po) at the indicated concentration for 16 h, and then stimulated with cGAMP for 3 h. **d** Immunoblot analysis of phosphorylation of TBK1. HEK293T cells expressing STING-S4D-AGIA were treated with cGAMP for 3 h and analyzed by immunoblot. **e** Effect of pomalidomide-induced STING degradation on the phosphorylation of TBK1. HEK293T cells expressing STING-S4D-AGIA were treated with cGAMP for 3 h and analyzed by immunoblot. **f** Quantitative RT-PCR of cGAMP-induced genes. HEK293T cells expressing STING-S4D-AGIA were treated with DM or Po for 16 h, and then stimulated with cGAMP for 3 h. The expression of IFN-β, CXCL10, CCL5, and ISG56 was measured by quantitative RT-PCR. The mRNA expression in cells expressing an empty vector was set to 1.0. Error bars in **a**, **c**, and **f** represent the mean ± SD (n = 3 or 5), and P values were calculated by one-way ANOVA with Tukey's post-hoc tests (NS not significant; *P < 0.05, **P < 0.01, ***P < 0.001, and ****P < 0.0001.

not promote the transcription of ISRE in HEK293T cells expressing an empty vector (Fig. 3a, bars 1–3). In HEK293T cells transiently expressing STING-AGIA or STING-S4D-AGIA, cGAMP treatment increased the luciferase activity (Fig. 3a, bars 4–5, and bars 7–8). Importantly, pomalidomide treatment suppressed the luciferase activity in HEK293T cells expressing STING-S4D-AGIA (Fig. 3a, bars 8–9) but not STING-AGIA (Fig. 3a, bars 5–6), and protein expression of STING-S4D-AGIA was also reduced by pomalidomide treatment (Fig. 3b). Furthermore, IMiD treatment suppressed the ISRE luciferase activity in a dose-dependent manner (Fig. 3c). After recognition of cGAMP by STING, the serine/threonine protein kinase TBK1 is activated by phosphorylation, resulting in dimerization and nuclear translocation of interferon regulatory factor 3 (IRF3)[32]. cGAMP treatment induced the phosphorylation of TBK1 in HEK293T cells expressing STING-S4D-AGIA (Fig. 3d). Therefore, we examined whether IMiD treatment suppressed the phosphorylation of TBK1 in cGAMP-stimulated HEK293T cells. Immunoblot analysis showed that TBK1 phosphorylation was

reduced by IMiD treatment (Fig. 3e). In addition, the expression of cGAMP-induced genes, including *IFNB1* (IFN-β), *CXCL10*, *CCL5*, and *ISG56*, was decreased by IMiD (Fig. 3f), indicating that the S4D system can be used to analyze target proteins in the type I interferon pathway. The S4D system is therefore a useful tool for analyzing the cellular functions of target proteins in transient expression experiments.

**Generation and evaluation of S4D-knock-in cells**. To evaluate IMiD-induced protein degradation, it would be ideal to integrate the tag into the genome, creating a system in which the tagged protein is endogenously expressed. However, it is unclear whether IMiD would induce degradation of endogenous tagged proteins[11]. Because NF-κB signal transduction is tightly regulated by many proteins through a very complicated signal transduction process[14–16], we focused on RelA, a component of the NF-κB transcription factor complex, and IκBα, an inhibitor of the nuclear translocation of NF-κB[12]. In addition, superfolder GFP (sfGFP)

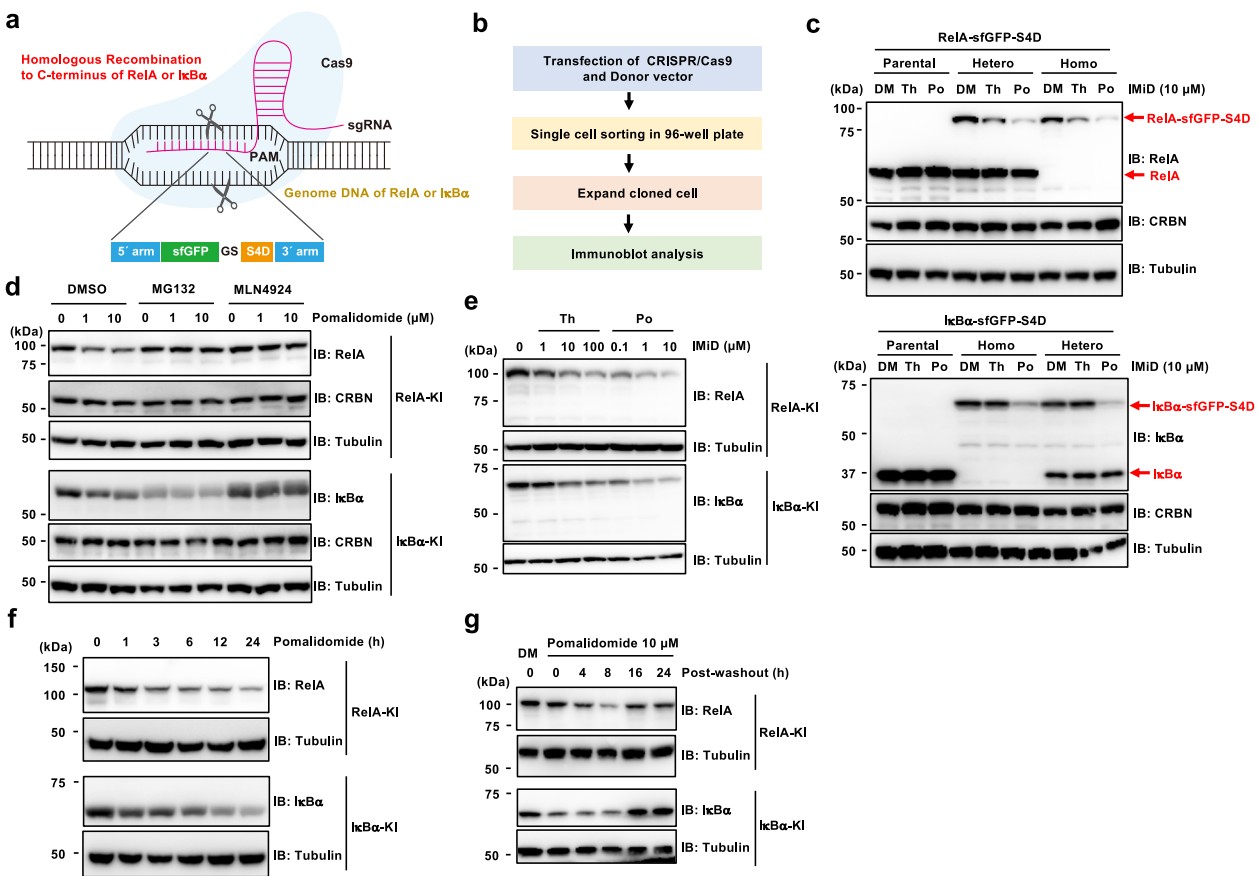

**Fig. 4 Generation and evaluation of knock-in (KI) cells expressing endogenous S4D-tagged protein. a** Schematic diagram of S4D tag insertion at the C-terminus by genome editing using the CRISPR/Cas9 system. **b** Flowchart of the generation of sfGFP-S4D-tagged KI clones in HeLa cells. **c** Immunoblot analysis of S4D-tagged endogenous RelA and IκBα in KI cells. Parental, heterozygous, or homozygous KI cells were treated with DMSO (DM), 10 μM thalidomide (Th), or 10 μM pomalidomide (Po) for 24 h. **d** Immunoblot analysis of RelA-sfGFP-S4D and IκBα-sfGFP-S4D in each KI cell treated with DM or Po in the presence of DM, MG132, or MLN4924 for 9 h. **e** Dose-dependent degradation of S4D-tagged endogenous RelA and IκBα. Each KI cell was treated with DM, Th, or Po at the indicated concentration for 24 h, and the lysates were analyzed by immunoblot. **f** Time-dependent degradation of S4D-tagged endogenous RelA and IκBα. Each KI cell was treated with Po for the indicated times, and the lysates were analyzed by immunoblot. **g** Analysis of the reversibility of protein degradation in the S4D system by immunoblot. After each KI cell was treated with DM or Po for 6 h, the culture medium was replaced, and cells were harvested after the indicated times.

was used for selection of cells with knocked-in genes. In a pilot study before the construction of knock-in (KI) cells, we constructed expression vectors encoding RelA-sfGFP-S4D or IκBα-sfGFP-S4D, as shown in Supplementary Fig. 5a, and examined whether these tagged proteins were degraded in the presence of IMiD using HEK293T-CRBN$^{-/-}$ cells expressing FLAG-CRBN. Immunoblot analysis showed that IMiD treatment induced the degradation of the S4D-tagged proteins (Supplementary Fig. 5b), and MLN4924 completely rescued the protein degradation (Supplementary Fig. 5c, lanes 1–4). In addition, CRBN-YW/AA, a mutant form of CRBN lacking the ability to bind to IMiDs[5], did not induce protein degradation (Supplementary Fig. 5c, lanes 5–8). As shown in Supplementary Fig. 5d and e, IMiD treatment also induced the degradation of both proteins in a dose- and time-dependent manner (Supplementary Fig. 5d and e).

Next, RelA-sfGFP-S4D and IκBα-sfGFP-S4D were overexpressed in HeLa cells, and protein degradation was evaluated. As in the HEK293T-CRBN$^{-/-}$ cells above, immunoblot analysis showed that endogenous CRBN decreased the expression of the two tagged proteins in the presence of IMiDs in HeLa cells (Supplementary Fig. 6). Then, we constructed a donor vector with homology arms and attempted to insert the S4D sequence into the C-terminus of the genes encoding RelA or IκBα in HeLa cells by genome editing using the clustered regularly interspersed

short palindromic repeats (CRISPR)/CRISPR-associated protein 9 (Cas9) system (Fig. 4a). GFP-positive cells in which the sequence had been successfully incorporated were isolated by FACS, and each cloned cell was expanded (Fig. 4b). As shown in Fig. 4c, the bands representing RelA and IκBα on immunoblots were shifted to a higher molecular weight (Fig. 4c) due to their fusion with sfGFP-S4D. Next, we investigated whether the endogenous S4D-tagged RelA and IκBα were degraded by IMiD treatment. As shown in Fig. 4c, the higher molecular weight band representing RelA- or IκBα-sfGFP-S4D was reduced by IMiD treatment, but the expression of untagged RelA and IκBα was not affected by IMiD treatment (Fig. 4c). Furthermore, the IMiD-induced protein degradation was not observed in the presence of MG132 or MLN4924 (Fig. 4d). In a previous study, the AID tag system was reported to be the most similar to the S4D tag system because it is comprised of a plant CRL1 E3 ligase complex and auxin[3]. AID-tagged proteins were degraded to below the basal level independently of auxin[33]. Therefore, in the development of novel chemical knockdown systems, it is important to consider the basal degradation of the tagged protein. As shown in Fig. 4c, the expression of S4D-tagged RelA and IκBα was decreased compared with the original levels of RelA and IκBα. However, MG132 and MLN4924 treatment did not increase their expression (Fig. 4d), demonstrating that the

S4D tag does not induce protein degradation to below the basal level.

We investigated the dose- and time-dependency of the S4D system on endogenous protein degradation. Thalidomide and pomalidomide treatments induced the degradation of S4D-tagged RelA and IκBα in a dose- and time-dependent manner (Fig. 4e, f). S4D-tagged endogenous RelA and IκBα were dramatically decreased by 3 h after supplementation with pomalidomide (Fig. 4f), indicating that the S4D system has the potential to quickly decrease the expression of endogenous tagged proteins. Because genomic approaches, such as knockout study by genome editing, do not have reversible effects on protein expression, it is possible that adaptation or bypassed signal transduction could occur in the cell clones. These unintended effects could complicate the proper analysis of the biological function of target proteins. One advantage of chemical knockdown using small molecules seems to be the reversibility of the protein degradation. We investigated the reversibility of protein degradation in the S4D system by washing out the pomalidomide. The protein expression of RelA- and IκBα-sfGFP-S4D was completely recovered 16 h after washing out the pomalidomide (Fig. 4g), showing that the S4D system has reversible effects on target protein expression. Taken together, these data demonstrate that the S4D system can reversibly induce the degradation of endogenous proteins of interest in a dose- and time-dependent manner.

**Functional analyses of RelA-S4D in KI cells.** TNF-α is widely known to induce the phosphorylation and degradation of IκBα, and the subsequent translocation of RelA to the nucleus[12]. To evaluate the potential artificial effects of sfGFP-S4D on NF-κB signal transduction, we examined whether TNF-α stimulation induces the nuclear translocation of RelA or the phosphorylation and degradation of IκBα in sfGFP-S4D-KI cells. Immunoblot analysis indicated that TNF-α induced the degradation of IκBα and the nuclear translocation of RelA in both KI cell lines to the same extent as in parental cells (Supplementary Fig. 7a). In addition, TNF-α-induced phosphorylation and degradation of the genome editing proteins were detected in both KI cell lines (Supplementary Fig. 7b), suggesting that the insertion of sfGFP-S4D into the genome did not cause artificial effects on NF-κB signal transduction in KI cells.

The effect of IMiD-dependent protein degradation of RelA on TNF-α-induced NF-κB activation and TNF-α-induced cell death was evaluated. During TNF-α stimulation, multiple proteins, including receptor interacting protein 1 (RIP1), TNFR1-associated death domain protein (TRADD), and the ubiquitin carboxyl-terminal hydrolase CYLD, are recruited to TNFR1, and these proteins form TNFR signaling complex I[34]. Then, NF-κB dimers consisting of RelA and p50 are activated and translocated to the nucleus, and NF-κB target genes are induced. As shown in Fig. 4f, because RelA-sfGFP-S4D was dramatically decreased 3–12 h after pomalidomide treatment, the homozygous KI cells of RelA-sfGFP-S4D-expressing were pretreated with pomalidomide for 12 h before TNF-α stimulation, and the expression of NF-κB target genes was examined. The expression of IκBα (NFKBIA) and IL-6 was increased by TNF-α stimulation in both parental and RelA-sfGFP-S4D-KI cells (Fig. 5a). Pomalidomide pretreatment significantly decreased the expression of these genes in RelA-sfGFP-S4D-KI cells (Fig. 5a) but did not do so in parental cells (Fig. 5a), indicating that pomalidomide-dependent degradation of RelA reduces NF-κB transcriptional activity. When TNFR signaling complex I fails to activate NF-κB, it transits to the formation of TNFR signaling complex II, which is comprised of RIP1, Fas-associated death domain protein (FADD), and

pro-caspase-8[35,36]. It has been reported that the formation of complex II induces apoptosis[35,36]. Consistent with this, as shown Supplementary Fig. 8, the combination of TNF-α and cyclohex-imide (CHX) remarkably decreased cell viability both of parental and RelA-sfGFP-S4D-KI cells[37]. Because IMiD treatment induced RelA degradation in the S4D-KI HeLa cells, it was predicted that the combination of TNF-α and IMiDs would result in TNF-α-induced apoptosis. Treatment with TNF-α or pomalidomide alone did not affect the viability of either parental or S4D-KI HeLa cells (Fig. 5b, top and middle panels). By contrast, TNF-α induced apoptosis in RelA-sfGFP-S4D-KI cells treated with pomalidomide in a dose-dependent manner (Fig. 5b, bottom panel, blue column). However, the combination of TNF-α and pomalidomide did not induce cell death in the parental cells (Fig. 5b, bottom panel, orange column), suggesting that this pomalidomide-dependent cell death results from the degradation of RelA by the S4D system. In addition, cell death was also observed by trypan blue staining (Fig. 5c). Immunoblot analysis showed cleavage of caspase-3, caspase-8, poly (ADP-ribose) polymerase (PARP), and RIP1 in a time-dependent manner (Fig. 5d). We next investigated whether the TNF-α- and pomalidomide-induced cell death is apoptotic cell death using zVAD-FMK, a pan-caspase inhibitor. Trypan blue staining and immunoblot analysis confirmed that the cell death was rescued by zVAD-FMK (Fig. 5e, f), indicating that the cell death observed in RelA-sfGFP-S4D-KI cells is TNF-α-induced apoptosis. From analyses of NF-κB transcriptional activity, we were therefore able to observe the predicted cellular events in response to degradation of RelA, demonstrating that the S4D system is a useful tool for understanding the functions of target proteins.

**IκBα degradation is insufficient for RelA nuclear translocation.** IκBα inhibits the nuclear translocation of NF-κB dimers in the canonical NF-κB pathway by masking the NLS of NF-κB[12]. NF-κB heterodimers consisting of RelA and p50 are a primary target of IκBα, and stimulus-dependent IκBα degradation leads to the nuclear localization of NF-κB[12]. It is thought that exposure of the NLS by degradation of IκBα triggers RelA:p50 heterodimers to translocate into the nucleus[12]. Therefore, we expected that pomalidomide treatment of IκBα-sfGFP-S4D-KI cells would induce the nuclear translocation of RelA:p50 heterodimers following IκBα degradation. However, surprisingly, RelA did not localize to the nucleus after pomalidomide treatment even though IκBα was degraded (Fig. 6a). Next, after pretreatment with pomalidomide, the localization of RelA:p50 after TNF-α stimulation was examined in parental and IκBα-sfGFP-S4D-KI cells. Consistent with the results shown in Fig. 6a, although RelA and p50 did not localize to the nucleus in the resting state (Fig. 6b, lane 1 and 4), TNF-α induced the nuclear localization of RelA and p50 (Fig. 6b, lanes 2–3 and 5–6), suggesting that pomalidomide treatment does not effect RelA nuclear translocation. Taken together, these results suggest that simple IκBα degradation cannot induce the nuclear translocation of RelA, and another factor is required.

**5-Hydroxythalidomide enhances selective degradation of the S4D system.** 5-Hydroxythalidomide (5-HT) (Fig. 7a) is a primary metabolite of thalidomide produced in several species cells[38,39], and we previously reported that 5-HT induced protein degradation of SALL4 more strongly than thalidomide[18]. On the other hand, 5-HT cannot induce protein degradation of IKZF1[18]. Therefore, we used 5-HT to bring out the potential of S4D system. To evaluate the efficiency of protein degradation by 5-HT, we produced HEK293T cells expressing S4D-FLuc-AGIA (S4D-FLuc), FLuc-S4D-AGIA (FLuc-S4D), or FLuc-I3D-AGIA

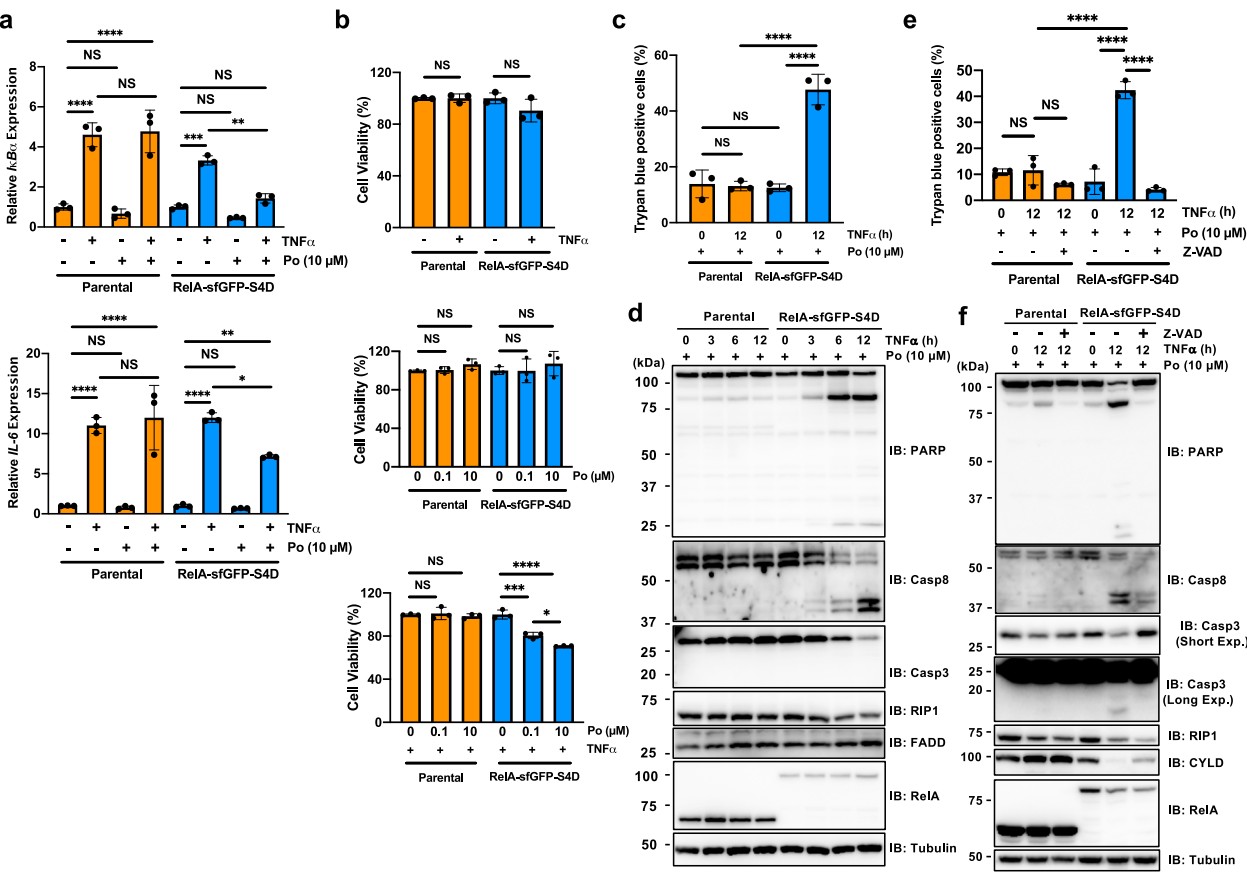

**Fig. 5 Analysis of RelA-dependent signaling in RelA-sfGFP-S4D-KI cells. a** Quantitative RT-PCR for the expression of TNF-α-induced genes. Parental or RelA-sfGFP-S4D-KI cells were pretreated with DMSO or pomalidomide (Po) for 24 h. Then, the cells were stimulated with 20 ng/ml TNF-α for 1 h, and the expression of IκBα or IL-6 was measured by quantitative RT-PCR. The mRNA expression in untreated parental HeLa cells was set to 1.0. **b, c** Pomalidomide causes TNF-α-induced cell death. Parental and RelA-sfGFP-S4D-KI cells pretreated with DMSO or Po for 12 h were stimulated with 20 ng/ml TNF-α for 12 h, and the viability was measured by MTS assay (**b**) or trypan blue staining (**c**). **d** Immunoblot analysis of pomalidomide-dependent TNF-α-induced cell death. Parental and RelA-sfGFP-S4D-KI cells pretreated with pomalidomide for 12 h were stimulated with 50 ng/ml TNF-α for the indicated times, and effectors of cell death were analyzed by immunoblot. **e, f** zVAD-FMK treatment rescued TNF-α-induced cell death in pomalidomide-treated KI cells. Parental and RelA-sfGFP-S4D-KI cells pretreated with 10 μM Po for 12 h were then treated with DMSO or 10 μM zVAD-FMK. After 2 h of zVAD-FMK treatment, the cells were stimulated with 50 ng/ml TNF-α for 12 h and the viability was measured by trypan blue staining (**e**), or effectors of apoptosis were analyzed by immunoblot (**f**). Error bars in **a–c** and **e** represent the mean ± SD ($n = 3$), and P values were calculated by one-way ANOVA with Tukey's post-hoc tests (NS not significant; $^{*}P < 0.05$, $^{**}P < 0.01$, $^{***}P < 0.001$, and $^{****}P < 0.0001$.

(FLuc-I3D) stably by using lentivirus (Fig. 7b). Luciferase assays showed that 5-HT more strongly provided degradation of both S4D-FLuc and FLuc-S4D than thalidomide whereas it did not induce protein degradation of FLuc-I3D (Fig. 7c). This result showed that 5-HT provides a highly specific degradation of S4D-tagged proteins but not I3D-tagged proteins. Furthermore, we investigated the efficiency of 5-HT-dependent protein degradation by dose-dependent experiments. Luciferase assay showed that 5-HT dramatically induced protein degradation of FLuc-S4D, compared with thalidomide (Fig. 7d), and the result was confirmed by immunoblot analysis (Fig. 7e). Furthermore, it was showed that 5-HT strongly induced protein degradation of both RelA-sfGFP-S4D and IκBα-sfGFP-S4D in KI cells (Fig. 7f). Importantly, the efficiency of protein degradation by 5-HT is similar to that of pomalidomide treatment (Fig. 7f). These results indicate that the combination of 5-HT and S4D tag enables the S4D system to be more powerful for analyzing the S4D-tagged target proteins with high specificity and efficiency.

## Discussion
IMiDs are drugs that act as "molecular glue," inducing the degradation of their target proteins by recruiting them to E3

ligases. In human cells, IMiDs-CRL4[CRBN] complex interacts with neosubstrates through a single zinc finger domain and induce the protein degradation[8–10]. Here, we developed an IMiD-induced SALL4 degron (S4D) composed of 28 amino acids (Fig. 1) and used it to analyze the biological functions of target proteins (Figs. 3, 5, and 6). Our results suggest that the S4D system can induce the proteasomal degradation of proteins localized to various cellular compartments in a dose- and time-dependent manner (Fig. 1 and 2). Because CRBN is ubiquitously expressed[5,40], we expect that a system based on the combination of IMiD and CRL4[CRBN] can be applied to the study of diverse proteins. First, the S4D system was used to analyze proteins of interest in transient expression experiments (Fig. 3). However, transient assay is limited to apply the cell lines lacking endogenous target protein because the endogenous target protein cannot be degradated by IMiD treatment. Therefore, it is required to select or to produce such deficient cell lines. To overcome this limitation, S4D-tagged proteins were introduced into the host cell genome using CRISPR/Cas9 and analyzed (Figs. 4, 5, and 6). Importantly, degradation of endogenously expressed target proteins in the KI cells was induced by IMiD treatment in a dose- and time-dependent manner (Fig. 4e, f). Recently, the IKZF3 degron (I3D) system based on

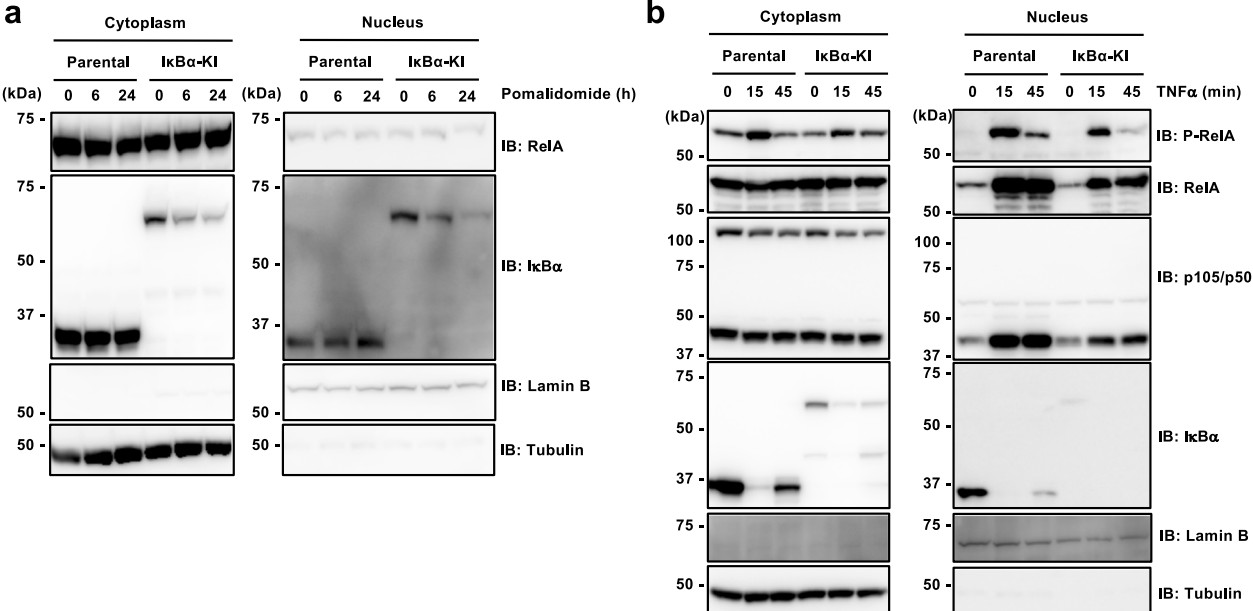

**Fig. 6 Analysis of IκBα-dependent signaling in IκBα-sfGFP-S4D-KI cells. a** Nuclear and cytoplasmic fractions were obtained from parental and IκBα-sfGFP-S4D-KI (IκBα-KI) cells after treatment with 10 μM pomalidomide for the indicated times and were analyzed by immunoblot. **b** Parental and IκBα-sfGFP-S4D-KI (IκBα-KI) cells were pretreated with 10 μM pomalidomide for 24 h, and then stimulated with TNF-α for the indicated times. Nuclear and cytoplasmic fractions were analyzed by immunoblot.

IMiD-CRBN was reported[11]. As shown in Supplementary Fig. 2b, the two tags induced almost the same level of protein degradation. However, although I3D did not induce degradation when it was expressed on the N-terminus of the target protein[11], the S4D system could be used as either an N- or a C-terminal tag (Fig. 2j, k, and Supplementary Fig. 2d). Furthermore, the S4D tag allowed for analysis of NF-κB signaling when it was used for endogenous tagged proteins, although this has not yet been shown for the I3D system. In addition, S4D tag system is acceptable to use 5-HT for the specific degradation (Fig. 7).

The combination of chemical and genetic methods is a powerful tool for analyzing the function of proteins, and many systems have been developed so far[1–4,11,33]. Compared with these other systems, the greatest advantage of the IMiD-CRBN system is that the tag is the shortest. In fact, 107 amino acids are required for the FKBP12-Shield-1 system[2], whereas the IKZF3 degron (25 amino acids)[11] and the SALL4 degron (28 amino acids) are substantially shorter. Because additional sequences may affect the function and localization of the target protein, a short tag is considered a great advantage. In addition, the AID system, a molecular glue in plant cells based on auxin, and TIR1, the auxin receptor in the Skp, cullin, F-box (SCF) E3 ligase complex, have also been used[3,33]. The greatest advantage of this system is that it is not necessary to consider the effects of endogenous proteolysis because there is no endogenous target protein in animal cells. However, this system requires co-expression of both the tagged protein and the TIR1 E3 ligase in animal cells. Additionally, the AID sequence is composed of at least 68 amino acids[33].

Proteolysis targeting chimera (PROTAC) is a powerful tool as another approach for protein knockdown using chemical compound[41–43]. PROTAC is produced as heterobifunctional molecules by fusion of two functional chemical compounds, which are E3 binder and target protein binder. PROTAC is being actively researched around the world because of the strong drug action that reduced expression of therapeutic target thoroughly. Furthermore, this concept also has powerful advantages that can target for almost proteins including undruggable proteins[41–43]. It is thought that PROTAC is a robust system for not only therapy

but also analyzing for biological function of target proteins[44]. The most notable advantage of PROTAC is that additional tag sequence need not to added to the target protein. Therefore, PROTAC is an ideal system in term of consideration of artificial effect of tag sequences. However, PROTAC is not available if there is not suitable target protein binder. In general, it is difficult to develop a novel chemical compound binding with target protein and it is important issue to have to consider the off-target of the novel binder. In contrast, the S4D system can be used for almost target proteins because required procedure is simple addition of the short tag sequence to target protein. Furthermore, thalidomide and its derivatives are most widely characterized molecular glue and their many neosubstrates have been already investigated. This is an important point to analyze whether the observed phenotypes are caused by degradation of the target protein or not. Taken together, the S4D system is a useful system in many situations.

Currently, it is thought that thalidomide derivatives have distinct specificity for neosubstrates[7,17]. For example, lenalidomide selectively induced protein degradation of casein kinase 1 alpha (CK1α)[7]. In addition, protein degradation of IKZF1 and IKZF3 was more strongly induced by lenalidomide and pomalidomide than thalidomide[6]. Actually, thalidomide-induced degradation was stronger using the S4D system than the I3D system (Supplementary Fig. 2b, Fig. 7c–e). Because it has been reported that thalidomide has fewer neosubstrates than lenalidomide and pomalidomide[9,10] and thalidomide is the most cost-effective in IMiDs, the use of thalidomide is an advantage. Recently, we reported that 5-HT, which is a thalidomide metabolite, has neosubstrate specificity between SALL4 and IKZF1[18]. In this study, we investigated whether neosubstrate specificity of IMiD can be applied the S4D system. As results, 5-HT more strongly induced protein degradation of S4D-tagged proteins than thalidomide (Fig. 7c–f). In contrast, I3D-tagged FLuc was not degraded by 5-HT treatment (Fig. 7c–e), showing that the neosubstrate specificity is conserved on short degron tag. Importantly, 5-HT can induce protein degradation of S4D-tagged protein at the same sensitivity with pomalidomide (Fig. 7f). Furthermore, lenalidomide and pomalidomide dramatically reduce protein expression of IKZF1

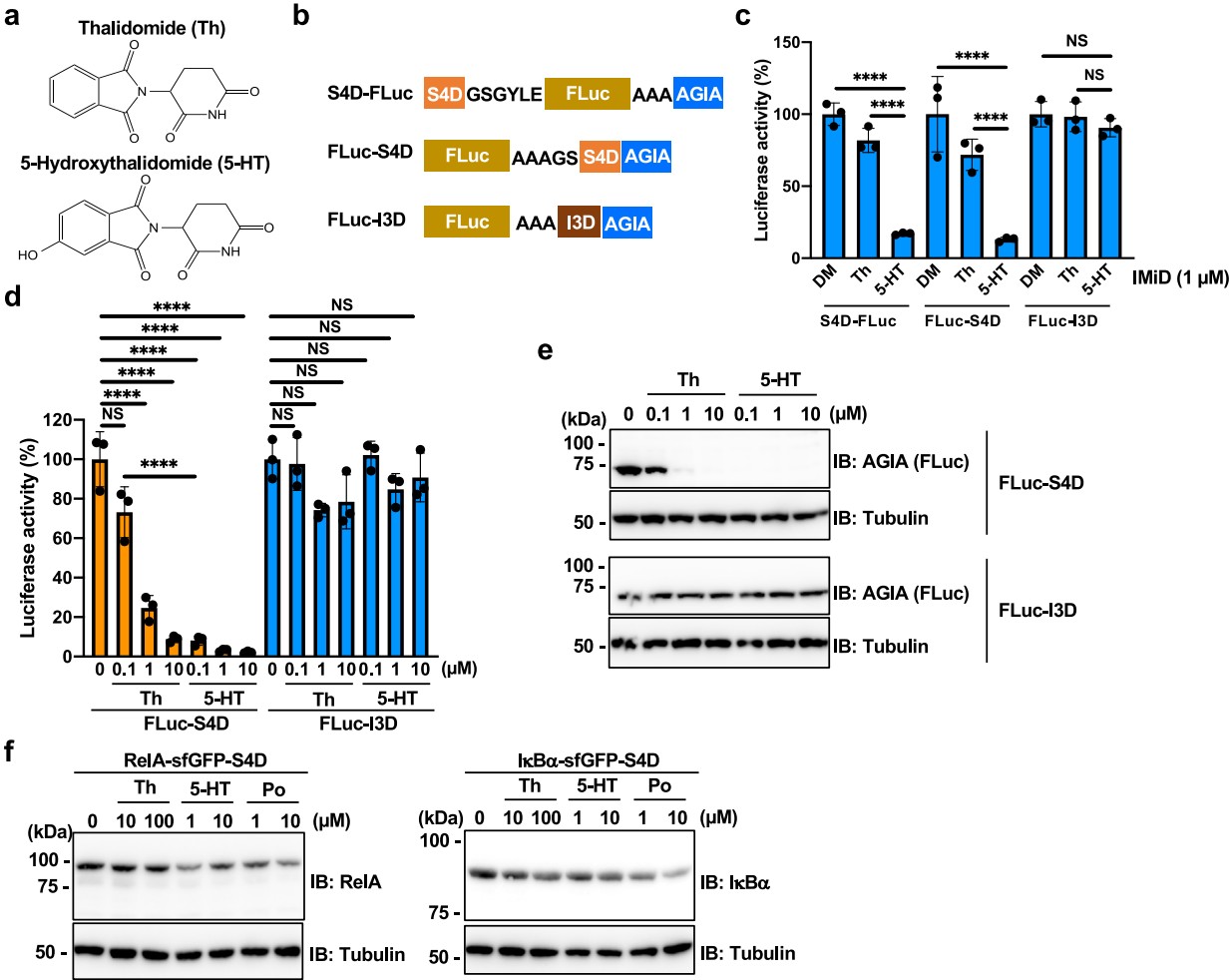

**Fig. 7 5-Hydroxythalidomide treatment strongly degradates the S4D-tagged proteins but not degradates the I3D-tagged proteins. a** Chemical structures of thalidomide (Th) and 5-hydroxythalidomide (5-HT). **b** Schematic diagram of the ORFs of S4D-FLuc-AGIA, FLuc-S4D-AGIA, and FLuc-I3D-AGIA in stable cell lines. **c** FLuc luciferase activity in lysates of HEK293T stable cells expressing S4D-FLuc-AGIA, FLuc-S4D-AGIA, or FLuc-I3D-AGIA treated with DMSO (DM), 1 μM thalidomide (Th), or 1 μM 5-hydroxythalidomide (5-HT) for 24 h. **d, e** Dose-dependent analyses of HEK293T stable cells expressing FLuc-S4D-AGIA or FLuc-I3D-AGIA treated with DM, Th, or 5-HT at the indicated concentration for 24 h. Luciferase activity analysis (**d**), immunoblot analysis (**e**), **f** Immunoblot analysis of RelA-sfGFP-S4D or IκBα-sfGFP-S4D in KI cells. Each KI cell was treated with DM, Th, 5-HT, or pomalidomide (Po) at the indicated concentration for 24 h. Error bars in **c** and **d** represent the mean ± SD (n = 3), and P values were calculated by one-way ANOVA with Tukey's post-hoc tests (NS not significant; and ****P < 0.0001.

and IKZF3 in several cell lines, such as multiple myeloma (MM)[6,7]. IKZF1 and IKZF3 are pivotal transcription factors for proliferation in MM cells. Therefore, the combination of S4D tag with 5-HT is a more ideal system if researchers use such cell lines or require the lower level of background degradation. In conclusion, we show that the S4D system is a powerful tool with significant advantages for the cell biology.

## Methods

**Reagents**. Thalidomide (Tokyo Chemical Industry), pomalidomide (Sigma–Aldrich), lenalidomide (FujiFilm Wako), 5-hydroxythalidomide (5-HT, synthesized as previously reported[45]), MG132 (Peptide Institute), MLN4924 (Chemscene), cycloheximide (Merck Millipore), and zVAD-FMK (Peptide Institute) were dissolved in DMSO (FujiFilm Wako) at 2–100 mM and stored at −20 °C as stock solutions. All drugs were diluted 1000-fold for in vivo experiments or 200-fold for in vitro experiments. 2′,3′-cGAMP (InvivoGen) was dissolved in endotoxin-free water. Recombinant human TNF-α (R&D Systems) was reconstituted according to the manufacturer's protocol.

**Antibodies**. The following horseradish peroxidase (HRP)-conjugated antibodies were used in this study: FLAG (Sigma–Aldrich, A8592, 1:5,000), AGIA[19] (produced in our laboratory, 1:10,000), α-tubulin (MBL, PM054-7, 1:10,000), and biotin (Cell Signaling Technology, #7075, 1:5,000). The following primary antibodies were

used in this study: CRBN (#71810, 1:1000), IκBα (#4814, 1:1000), phospho-IκBα (#9246, 1:1000), NF-κB1 p105/p50 (#12540, 1:1000), p65 (RelA) (#8242, 1:1000), phospho-p65 (RelA) (#3033, 1:1000), CYLD (#8462, 1:1000), RIP1 (#3493, 1:1000), caspase-8 (#9746, 1:1000), caspase-3 (#9662, 1:1000), PARP (#9542, 1:1000), FADD (#2782, 1:1000), phospho-TBK1 (#5483, 1:1000), and TBK1 (#3504, 1:1000) (all from Cell Signaling Technology); and lamin B (#sc-6217, 1:500) (from Santa Cruz Biotechnology). Anti-rabbit IgG (HRP-conjugated, Cell Signaling Technology, #7074, 1:10,000), anti-mouse IgG (HRP-conjugated, Cell Signaling Technology, #7076, 1:10,000), and anti-goat IgG (HRP-conjugated, Invitrogen/Thermo Fisher Scientific, #81-1620, 1:10,000) were used as secondary antibodies.

**Immunoblot analysis**. Protein samples were separated by SDS-PAGE and transferred to polyvinylidene difluoride (PVDF) membranes (Millipore). The membranes were blocked using 5% skim milk (Megmilk Snow Brand) in TBST [20 mM Tris-HCl (pH 7.5), 150 mM NaCl, 0.05% Tween20] at room temperature for 1 h, and then treated with the appropriate antibodies. Immobilon (Millipore) or Immuno-Star LD (FujiFilm Wako Pure Chemical Corporation) was used as a substrate for HRP, and the luminescent signal was detected on an ImageQuant LAS 4000 mini (GE Healthcare). In some blots, the membrane was stripped with Stripping Solution (FujiFilm Wako Pure Chemical Corporation) and reprobed with other antibodies.

**Cell culture and transfection**. HEK293T or HeLa cells were cultured in low-glucose DMEM (FujiFilm Wako Pure Chemical Corporation) supplemented with 10% fetal bovine serum (FujiFilm Wako Pure Chemical Corporation), 100 U/ml

penicillin, and 100 μg/ml streptomycin (Gibco/Thermo Fisher Scientific) at 37 °C under 5% $CO_2$. HEK293T or HeLa cells were transfected using *TransIT*-LT1 transfection reagent (Mirus Bio) or Polyethyleneimine (PEI) Max (MW 40,000) (PolyScience, Inc.).

**Construction of CRBN-KO HEK293T cells**. CRBN-KO HEK293T cells were generated by genome editing using CRISPR/CAS9 system. The guide nucleotide sequence 5′-ACTCCGGGCGGTTACCAGGC- 3′ in the human *CRBN* gene was inserted into the Guide-It plasmid vector (Takara Bio). HEK293T cells were cultured in six-well plates and transfected with the plasmid for 2 days. Then, GFP-positive cells were sorted on a FACSAria (BD Biosciences), and cell clones were obtained by limiting dilution. Genomic DNA was then isolated, and the mutation was confirmed by sequencing after TA cloning (Toyobo).

**Plasmids**. The pDONR221, pcDNA3.1(+), and pCAGGS plasmids were purchased from Invitrogen or Riken, and the pEU vector for the wheat cell-free system was constructed in our laboratory, as previously described[46]. The pcDNA3.1 (+)-FLAG-GW, pcDNA3.1(+)-AGIA-MCS, pEU-bls-GW, pEU-FLAG-GST-MCS, pCAGGS-MCS-S4D-AGIA, pCAGGS-MCS-(IKZF3 degron)-AGIA, and pCAGGS-S4D-MCS-AGIA plasmids were constructed by PCR and restriction enzymes. The SALL4 gene sequence was purchased from the Kazusa DNA Research Institute. The Venus[19] and SALL4 sequences were amplified, and restriction enzyme sites were added by PCR and cloned into pEU-FLAG-GST-MCS. The pEU-FLAG-GST-Venus-m1, -m2, and -m3 plasmids were constructed by inverse PCR from the pEU-FLAG-GST-Venus-SALL4 plasmid. Venus-m1, -m2, and -m3 were digested by restriction enzymes from pEU-FLAG-GST-Venus-m1, -m2, and -m3, and cloned into pcDNA3.1(+)-AGIA-MCS. Venus was also cloned into the pCAGGS-MCS-S4D-AGIA, pCAGGS-S4D-MCS-AGIA, and pCAGGS-MCS-IKZF3 degron (I3D)-AGIA plasmids using restriction enzymes. The open reading frames (ORFs) of *CRBN*, *RELA* (RelA), *NFKBIA* (IκBα), *DRD1*, and *TP53* (p53) were purchased from the Mammalian Gene Collection (MGC)[47], while the ORF of STING was purchased from Promega (Flexi Clone). The ORF of *MAVS* was provided by Dr. Youichi Suzuki. CRBN was amplified, and the BP reaction sequence (attB and attP) was added by PCR and cloned into pDONR221 using BP recombination (Invitrogen/Thermo Fisher Scientific). Then, pDONR221-CRBN was recombined into pEU-bls-GW or pcDNA3.1(+)-FLAG-GW using LR recombination (attL and attR). RelA, IκBα, and sfGFP were amplified, and restriction enzyme sites were added by PCR. Then, RelA-sfGFP and IκBα-sfGFP were cloned into pcDNA3.1 (+)-AGIA-MCS, and the S4D tag was added to the plasmids pcDNA3.1(+)-AGIA-RelA-sfGFP and pcDNA3.1(+)-AGIA-IκBα-sfGFP using inverse PCR. p53, DRD1, STING, and MAVS were amplified, and restriction enzyme sites were added by PCR. They were then cloned into pCAGGS-MCS-AGIA, pCAGGS-MCS-S4D-AGIA, or pCAGGS-S4D-MCS-AGIA. Firefly luciferase (FLuc) was amplified by PCR using pGL4.32[luc2P/NF-kB-RE/Hygro] as a template and cloned into pCAGGS-MCS-AGIA or pCAGGS-MCS-S4D-AGIA using restriction enzymes.

For the generation of KI cells, genomic DNA was isolated from HeLa cells, locus-specific 5′ and 3′ homology arms were amplified by PCR, and restriction enzyme sites were added. These homology arms and protospacer adjacent motif (PAM) sequences were designed to remove the stop codon from the gene ORF, and to attach sfGFP and the S4D tag. Then, these fragments were cloned into the LITMUS29 vector[48] using the In-Fusion system (Takara Bio). The sgRNA vector (Addgene) was used to produce KI cells. The guide nucleotide sequences were as follows: 5′-AGTCAGATCAGCTCCTAAGG-3′ (RelA) and 5′-GCAAAGGGGCTG AAAGAACA- 3′ (IκBα). The target sequences were cloned into the sgRNA vectors using Gibson assembly (New England Biolabs).

For cells stably expressing DRD1 or FLuc, pCSII-CMV-DRD1-S4D-AGIA, FLuc-AGIA, -FLuc-S4D-AGIA, or -FLuc-I3D-AGIA was constructed by using PCR and restriction enzyme.

**Produce of lentivirus**. Lentivirus for expressing DRD1-S4D-AGIA, S4D-FLuc-AGIA, FLuc-S4D-AGIA, or FLuc-I3D-AGIA were produced in HEK293T cells by transfection of pCSII-CMV-DRD1 or -FLuc expression vector together with pCMV-VSV-G-RSV-Rev and pCAG-HIVgp. Culture medium was exchanged after 24 h transfection, and the cells were cultured for 48 h. Then, the lentiviruses were concentrated by using Lenti-X concentrator (Takara Bio).

**Generation of stably HEK293T or HeLa cells lines**. HEK293T or HeLa cells supplemented with 5 μg/ml Polybrene (Nacalai Tesque) were infected by the appropriate lentivirus, and culture medium was exchanged after 24 h infection. 5 μg/ml blasticidin S (InvivoGen) selection was started after 24 h from exchange of culture medium.

**Synthesis of recombinant protein using the wheat cell-free system**. In vitro transcription and wheat cell-free protein synthesis were performed using the WEPRO1240 expression kit (Cell-Free Sciences). Transcription was performed using SP6 RNA polymerase with the plasmids or DNA fragments described above as templates. The translation reaction was performed in bilayer mode using the WEPRO1240 expression kit (Cell-Free Sciences), according to the manufacturer's

protocol. For biotin labeling, cell-free synthesized crude biotin ligase (BirA) produced using the wheat cell-free expression system was added to the bottom layer, and 0.5 μM (final concentration) of d-biotin (Nacalai Tesque) was added to both the upper and lower layers, as described previously[49].

**AlphaScreen-based interaction assay of CRBN-Venus-S4D**. IMiD at a final concentration of 50 μM and 0.5 μl biotinylated CRBN were mixed in 15 μl AlphaScreen buffer containing 100 mM Tris (pH 8.0), 0.01% Tween20, 100 mM NaCl, and 1 mg/ml BSA. Then, 5 μl of a mixture containing 0.8 μl FLAG-GST-Venus-m1, -m2, or -m3 in AlphaScreen buffer was added, and the reaction mixtures were incubated at 26 °C for 1 h in a 384-well AlphaPlate (PerkinElmer). Subsequently, 5 μl of a detection mixture containing 0.2 μg/ml anti-DYKDDDDK mouse mAb (Wako), 0.08 μl streptavidin-coated donor beads, and 0.08 μl Protein A-coated acceptor beads (PerkinElmer) in AlphaScreen buffer was added to each well. After incubation at 26 °C for 1 h, luminescence was read on an EnVision plate reader (PerkinElmer).

**In-cell IMiD-dependent degradation of S4D-tagged proteins**. To confirm IMiD-dependent degradation of Venus-tagged proteins, HEK293T-CRBN$^{−/−}$ cells were cultured in 24-well plates and transfected with 400 ng pcDNA3.1(+)-FLAG-CRBN-WT together with 50 ng pcDNA3.1(+)-AGIA-Venus-m1, -m2, -m3, or -m2 variant (G416A). After the cells were transfected for 6 h, they were treated with IMiD or DMSO (0.1%) in culture medium, at the times and concentrations indicated in each figure.

For degradation Venus-S4D by endogenous CRBN, HEK293T cells were cultured in 24-well plates and transfected with 5 ng pcDNA3.1(+)-AGIA-Venus-S4D. After the cells were transfected for 6 h, they were treated with IMiD or DMSO (0.1%) in culture medium, at the times and concentrations indicated in each figure.

For degradation of S4D-Venus, Venus-S4D, and Venus-I3D, HEK293T cells were cultured in 48-well plates and transfected with 10 ng pCAGGS-S4D-Venus-AGIA, pCAGGS-Venus-S4D-AGIA, or pCAGGS-Venus-I3D-AGIA. After the cells were transfected for 6 h, they were treated with IMiD or DMSO (0.1%) in culture medium, at the times and concentrations indicated in each figure.

For degradation of FLuc-S4D, HEK293T cells were cultured in 48-well plates and transfected with 2 ng pCAGGS-FLuc-AGIA or 25 ng pCAGGS-FLuc-S4D-AGIA. After the cells were transfected for 6 h, they were treated with IMiD or DMSO (0.1%) in culture medium, at the times and concentrations indicated in each figure.

To examine the degradation of proteins with various subcellular localizations, we cultured HEK293T cells, HeLa cells, or HEK293T-CRBN$^{−/−}$ cells in 48-well plates and transfected with 200 ng pcDNA3.1(+)-FLAG-CRBN-WT or 200 ng pcDNA3.1(+)-empty together with 30 ng pCAGGS-p53-S4D-AGIA, 20 ng pCAGGS-STING-S4D-AGIA, 10 ng pCAGGS-DRD1-S4D-AGIA, 100 ng pCAGGS-MAVS-S4D-AGIA, 200 ng pCAGGS-GM130-S4D-AGIA. After the cells were transfected for 8 h, they were treated with IMiD or DMSO (0.1%) in culture medium, at the times and concentrations indicated in each figure.

For degradation of DRD1-S4D-AGIA in stable cell line, HeLa cells expressing DRD1-S4D-AGIA were cultured in 24-well plates. Then, the cells were treated with IMiD or DMSO (0.1%) in culture medium, at the times and concentrations indicated in each figure.

To examine the degradation of S4D-tagged RelA or IκBα, HeLa cells or HEK293T-CRBN$^{−/−}$ cells were cultured in 48-well plates and transfected with 200 ng pcDNA3.1(+)-FLAG-CRBN-WT or 200 ng pcDNA3.1(+)-FLAG-CRBN-YW/ AA together with 15 ng pcDNA3.1(+)-AGIA-sfGFP-RelA or 15 ng pCDNA3.1 (+)-AGIA-sfGFP-IκBα. After the cells were transfected for 8 h, they were treated with IMiD or DMSO (0.1%) in culture medium, at the times and concentrations indicated in each figure.

To examine the degradation of endogenous S4D-tagged RelA or IκBα, parental, heterogeneous, or homogeneous KI HeLa cells were cultured in 48-well plates. Then, the cells were treated with IMiD or DMSO (0.1%) in culture medium, at the times and concentrations indicated in each figure.

To show that IMiD-dependent degradation of S4D-tagged proteins is dependent on CRL and the 26 S proteasome, the cells were treated with 2 μM MLN4924 and 10 μM MG132 (0.2% DMSO) for the times indicated in each figure.

For protein degradation by 5-HT, HEK293T cells expressing S4D-FLuc-AGIA, FLuc-S4D-AGIA, or FLuc-I3D-AGIA or KI HeLa cells were culture in 48-well plates. Then, the cells were treated with IMiD or DMSO (0.1%) in culture medium, at the times and concentrations indicated in each figure.

The cells were lysed by boiling at 95 °C for 5 min in 1× sample buffer containing 5% 2-mercaptoethanol, and the lysates were analyzed by immunoblot. In the case of cells expressing transmembrane protein, the cells were lysed by boiling at 50 °C for 10 min in 1× sample buffer containing 5% 2-mercaptoethanol, and the lysates were analyzed by immunoblot after sonication.

**Quantitative evaluation using Firefly luciferase activity**. HEK293T cells were cultured in 96-well plates and transfected with 0.1 ng pCAGGS-FLuc-AGIA or 1 ng pCAGGS-FLuc-S4D-AGIA. After the cells were transfected for 6 h, they were treated with IMiD or DMSO (0.1%) in culture medium, at the times and concentrations indicated in each figure. In the case of stable cell lines expressing

S4D-FLuc-AGIA, FLuc-S4D-AGIA, or FLuc-I3D-AGIA, the cells were cultured in 96-well plates and were treated with IMiD or DMSO (0.1%) in culture medium, at the times and concentrations indicated in each figure. The cells were lysed with 45 μl 1× Passive Lysis Buffer (Promega), and the lysates were diluted 200-fold with 1× Passive Lysis Buffer. Then, FLuc luciferase activity was measured on a Glomax luminometer (Promega) using the luciferase assay system (Promega).

**Immunofluorescence staining.** HeLa cells were cultured on 12 mm poly-L-lysine-coated glass slides (Sigma–Aldrich) in 24-well plates and transfected with 200 ng pCAGGS-p53-S4D-AGIA, 100 ng pCAGGS-STING-S4D-AGIA, 400 ng pCAGGS-DRD1-S4D-AGIA, 500 ng pCAGGS-MAVS-S4D-AGIA, or 300 ng pCAGGS-GM130-S4D-AGIA. After transfection for 24 h, the cells were fixed with 4% paraformaldehyde (PFA) in PBS at room temperature for 15 min, and permeabilization was performed with 0.1% Triton X-100 in PBS (for p53, STING, MAVS, and GM130) or 0.01% digitonin in PBS (for DRD1) at room temperature for 15 min. Then, the cells were incubated with 0.5% cattle serum (CS) in TBST at 4 °C for 1 h, and incubated with an anti-AGIA antibody (produced by our laboratory) at 4 °C for 16 h. After washing with TBST at room temperature for 15 min, the cells were incubated with an Alexa Flour 488-conjugated anti-rabbit secondary antibody at room temperature for 1 h. The nucleus, ER, and mitochondria were stained with 4′,6-diamidino-2-phenylindole (DAPI), ER-ID (Enzo), or Mito-ID (Enzo), respectively, according to the manufacturer's protocol. After washing with TBST at room temperature for 15 min, the stained cells were mounted with anti-fade (Invitrogen/Thermo Fisher Scientific) and observed under a BZ-X810 Microscope (Keyence).

For immunofluorescence staining of GM130, the cells were incubated with an anti-AGIA antibody (for GM130-S4D-AGIA) together with an anti-GM130 antibody (for staining Golgi apparatus) (MBL, M179-3MS, 1:200) at 4 °C for 16 h. After washing with TBST at room temperature for 15 min, the cells were incubated with an Alexa Flour 488-conjugated anti-rabbit secondary antibody and Alexa Flour 555-conjugated anti-mouse secondary antibody at room temperature for 1 h.

For immunofluorescence staining of stably expressed DRD1, HeLa cells expressing DRD1-S4D-AGIA were incubated with an anti-AGIA antibody[19] (produced in our laboratory, 1:1000) at 4 °C for 16 h. After washing with TBST at room temperature for 15 min, the cells were incubated with an Alexa Flour 488-conjugated anti-rabbit secondary antibody at room temperature for 1 h.

**cGAMP-induced ISRE luciferase reporter assay.** HEK293T cells were cultured in 96-well plates and transfected with 1 ng empty vector, 0.1 ng pCAGGS-STING-AGIA, or 1 ng pCAGGS-STING-S4D-AGIA together with 4 ng pGL4-ISRE-promoter-Luc plasmid (Promega) and 1 ng pRL-TK-Renilla-Luc plasmid (Promega). After the cells were transfected for 6 h, they were pretreated with DMSO (0.1%) or pomalidomide in culture medium for 16 h. The cells were stimulated with 2 μg/ml cGAMP in permeabilization buffer [50 mM HEPES (pH 7.2–7.5), 100 mM KCl, 3 mM MgCl2, 0.1 mM dithiothreitol (DTT), 85 mM sucrose, 1 mM ATP, 0.2% BSA, 10 μg/ml digitonin] for 3 h, and then lysed with 45 μl 1× Passive Lysis Buffer, and the Firefly and Renilla luciferase activity was measured on a Glomax luminometer using the dual-luciferase reporter assay system (Promega).

**Generation of KI cells expressing RelA- or IκBα-sfGFP-S4D.** HeLa cells were cultured in six-well plates for 24 h and we made RelA-sfGFP-S4D and IkBa-sfGFP-S4D-KI cells as described previously[48] by using *Trans*IT-LT1 transfection reagent (Mirus Bio). After 2 days of transfection, the cells were passaged and cultured in 10 cm dishes. After 2 days of culture, the GFP-positive cells were isolated by sorting on a FACSAria (BD Biosciences). The insertion of sfGFP-S4D into RelA or IκBα gene loci was confirmed by a band shift on immunoblot analysis.

**Reversibility of the S4D system.** To examine the reversibility of the S4D system, KI (RelA-sfGFP-S4D or IκBα-sfGFP-S4D) HeLa cells were cultured in 12-well plates. The cells were treated with DMSO or 10 μM pomalidomide for 6 h, and then the DMSO or pomalidomide was removed by washing with culture medium. Cells were harvested at the times indicated in Fig. 4g and lysed with RIPA buffer containing protease inhibitor cocktail (Sigma–Aldrich). The protein concentrations of the lysates were quantitated by bicinchoninic acid (BCA) assay (Thermo Fisher Scientific), and the expression of RelA-sfGFP-S4D or IκBα-sfGFP-S4D was analyzed by immunoblot.

**Quantitative RT-PCR.** For analysis of cGAMP-induced genes, HEK293T cells were cultured in 96-well plates and transfected with 1 ng pCAGGS-STING-S4D-AGIA. After the cells were transfected for 6 h, they were pretreated with DMSO (0.1%) or 10 μM pomalidomide in culture medium for 16 h. Then, the cells were stimulated with 2 μg/ml cGAMP in permeabilization buffer. After 3 h, total RNA was isolated from the cells using the SuperPrep cell lysis kit (Toyobo), and cDNA was synthesized using the SuperPrep RT kit (Toyobo), according to the manufacturer's protocol. RT-PCR was performed using KOD SYBR qPCR Mix (Toyobo), and data were normalized against the GAPDH mRNA levels. PCR primers are as follows[50,51]: IFN-β sense 5′-GGACCATAGTCAGAGTGGAA ATCCTAAG-3′, IFN-β anti-sense 5′-CACTTAAACAGCATCTGCTGGTTGA AG-3′, CXCL10 sense 5′-AGCAGAGGAACCTCCAGTCT-3′, CXCL10 anti-sense

5′-AGGTACTCCTTGAATGCCACT-3′, CCL5 sense 5′-CTGCTTTGCCTACAT TGCCC-3′, CCL5 anti-sense 5′-TCGGGTGACAAAGACGACTG-3′, ISG56 sense 5′- CAAAGGGCAAAACGAGGCAG-3′, ISG56 anti-sense 5′-CCCAG GCATA GTTTCCCCAG-3′, GAPDH sense 5′-AGCAACAGGGTGGTGGAC-3′, and GAPDH anti-sense 5′-GTGTGGTGGGGGACTGAG-3′.

For analysis of TNF-α-induced genes, parental or KI (RelA-sfGFP-S4D) HeLa cells were cultured in 96-well plates and pretreated with DMSO (0.1%) or 10 μM pomalidomide. After 12 h of pomalidomide pretreatment, the cells were treated with 20 ng/ml TNF-α for 2 h. Total RNA was isolated from the cells using the SuperPrep cell lysis kit (Toyobo), and cDNA was synthesized using the SuperPrep RT kit (Toyobo), according to the manufacturer's protocols. RT-PCR was performed using KOD SYBR qPCR Mix (Toyobo), and data were normalized against GAPDH mRNA levels. PCR primers are as follows[37]: IκBα sense 5′-CGGGCTGAAGAAGGAGCGGC-3′, IκBα anti-sense 5′-ACGAGTCCCC GTCCTCGGTG-3′, IL-6 sense 5′-AGCCACTCACCTCTTCAGAAC-3′, and IL-6 anti-sense 5′-GCCTCTTTGCTGCTTTCACAC-3′.

**Extraction of nuclei and cytoplasm.** Parental or KI HeLa cells were cultured in 6-well or 12-well plates and pretreated with DMSO (0.1%) or pomalidomide in culture medium for the times indicated in each figure. Then, the cells were stimulated with 20 ng/ml TNF-α for the indicated times and harvested using TrypLE (Gibco/Thermo Fisher Scientific). Nuclear and cytoplasmic fractions were extracted using the NE-PER Nuclear and Cytoplasmic Kit (Thermo Fisher Scientific), according to the manufacturer's protocol. The protein fractions were separated by SDS-PAGE and analyzed by immunoblot using specific antibodies.

**Cell viability assay.** For combination treatment with TNF-α and cycloheximide (CHX), parental or KI (RelA-sfGFP-S4D) HeLa cells were cultured in 96-well plates and treated with 20 ng/ml TNF-α and 10 μg/ml CHX for 24 h. For combination treatment with TNF-α and pomalidomide, parental or KI (RelA-sfGFP-S4D) HeLa cells were cultured in 96-well plates and pretreated with DMSO (0.1%) or pomalidomide in culture medium. After 12 h of pomalidomide treatment, the cells were treated with 20 ng/ml TNF-α for 12 h. Then, the number of viable cells was measured using the CellTiter 96 AQueous One Solution Cell Proliferation Assay kit (Promega) and a SpecraMax M3 microplate reader (Molecular Devices).

**Analysis of TNF-α-induced cell death by immunoblot.** Parental or KI (RelA-sfGFP-S4D) HeLa cells were cultured in six-well plates and pretreated with pomalidomide at the concentrations indicated in each figure. After 12 h of pomalidomide pretreatment, cells were treated with 50 ng/ml TNF-α for the times indicated in each figure. Then, the cells were harvested using TrypLE, and the number of dead cells was counted by trypan blue staining using a Countess Automated Cell Counter (Invitrogen/Thermo Fisher Scientific). The cells were lysed with RIPA buffer containing protease inhibitor cocktail (Sigma–Aldrich), and protein concentrations were measured by BCA assay (Thermo Scientific). Then, the lysates were analyzed by immunoblot with specific antibodies. For experiments with zVAD-FMK, the cells were treated with DMSO (0.1%) or 10 μM zVAD-FMK for 2 h before stimulation with TNF-α.

**Statistics and reproducibility.** All data were analyzed from at least three technical repeats ($n = 3–5$) and were presented as the means ± standard deviation (SD). Significant changes were analyzed by one-way analysis of variants (ANOVA) followed by Tukey's tests using GraphPad Prism (version 8) software (GraphPad, Inc.). Western blots and immunofluorescence images were repeated more than three times with similar results.

**Reporting summary.** Further information on research design is available in the Nature Research Reporting Summary linked to this article.

## Data availability
Source data for all graphs in this article are included in Supplementary Data 1. Uncropped data for all blots in this article are included Supplementary Information. The information and data in this article are available from the corresponding author on reasonable request.

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

## Acknowledgements

We thank C. Takahashi and C. Furukawa for technical assistance, and the Applied Protein Research Laboratory of Ehime University. We also thank Prof. H. Kosako (Tokushima University) for advices about proteins with different subcellular localizations. This work was mainly supported by the Platform Project for Supporting Drug Discovery and Life Science Research (Basis for Supporting Innovative Drug Discovery and Life Science Research (BINDS)) from AMED under Grant Number JP19am0101077 (T.S.), a Grant-in-Aid for Scientific Research on Innovative Areas (JP16H06579 for T.S.) from the Japan Society for the Promotion of Science (JSPS). This work was also partially supported by JSPS KAKENHI (JP17J08477 for S.Y., JP16H04729, JP19H03218 for T.S., 18K08574 for H.N.-F., JP17H06112 for N.S.), a Grant-in-Aid for JSPS Research Fellow (JP17J08477 for S.Y.) from JSPS, and Takeda Science Foundation.

## Author contributions

S.Y. and Y.S. performed the biochemical, molecular, and cellular biology experiments. S.M. performed vector constructions and AlphaScreen experiments. H.N.-F. designed and supported for the knock-in study. N.S. synthesized and analyzed the 5-hydroxythalidomide. S.Y. and T.S. analyzed the data, designed the study, wrote the paper, and all authors contributed to the manuscript.

## Competing interests

The authors declare no competing interests.
