## [Peer Review File · Communications Biology]

Reviewers' comments:

Reviewer #1 (Remarks to the Author):

The authors introduce another method for targeted protein degradation using a SAL4 ZnF degron protein tag fusion and IMiDs. This closely resembles previous work using an IKZF3 tag, but will be of new interest due to its success in N-terminal fusions and a variety of substrates with diverse subcellular localisation. Specific comments are given below.

Results text

Line 73. The authors should state that they are co-transfecting CRBN to allow degradation in these cells otherwise the results as stated don't make sense (I don't see why the readers should hunt through the figure legend to find basic points on the experimental hypotheses).

Line 80. Typo MNL4924

Line 133. "...suggesting that the degradation of DRD1 occurs on the plasma membrane". If internalised the DRD1 N-terminus would still be inside endosomes or lysosomes that may be inaccessible to CRBN. Thus, I am not sure the statement that DRD1 degradation must happen at the plasma membrane is strictly true.

Line 159. "The S4D system is therefore a useful tool for analyzing the cellular functions of target proteins in transient expression experiments". In practise, the authors example is greatly aided by the having cells that lack endogenous expression of the target protein.

Figures

The font size used for text within the figures is far far too small throughout (it is a struggle to read – e.g. especially fig 5d label "pretreated pomalidomide" needed 200% zoom to read).

Fig 1C legend. Abbreviation "DM" should be noted as used for DMSO. Why is CRBN band poorly seen in DMSO or thalidomide-treated cells? It is not ideal to judge neosubstrate levels when the control levels of E3 are so varied.

Fig2D. Why does POM stabilise MAVS in CRBN^{-/-} cells? Is POM binding to the SAL4 tag helping to give a thermostabilisation effect to the fusion protein? Is this a sign of side effects from the S4D system that future users should be made aware of?

Methods

Plasmids section. Small differences in construct design appear to effect the ability of the S4D system to work as judged in Fig 1c by the failure of the m3 degron motif. The authors do not describe the restriction sites employed or if spacer amino acids are introduced (e.g. small Gly, Ala or Ser type a.a.) between tag and fusion target protein. I can imagine these elements might effect the outcome. As such, it is impossible for any reader to reproduce the constructs described in this paper. Thus, further details are required. Best would be a supplemental figure showing the cloning map. Otherwise the methods appear quite detailed.

Discussion.

The authors mention one advantage of their system is the efficacy with thalidomide versus the IKZF3 tag. However, the S4D tag results suggest that 10 micromolar pomalidomide is the only treatment that gives "near" complete knockdown, whereas thalidomide is the least effective. I imagine the authors would like their system to be widely adopted, used for other publications and cited. Therefore,

the authors should comment on their recommendations to future users of the conditions they recommend (choice of IMiD, treatment time etc). They should comment on the specificity issues of using 10 micromolar pomalidomide.

Reviewer #2 (Remarks to the Author):

IMiDs such as thalidomide and its derivatives have been shown to regulate CRBN-ubiquitin ligase dependent (neo)substrate degradation, e.g., IKZF3 and Sal-like protein 4 (SALL4). This manuscript reports development of a Sal-like protein 4 (SALL4) degron (S4D) system, a binding interface for CRBN ubiquitin ligase in presence of IMiDs, for the use of protein substrate degradation via Cullin/UPS. The manuscript shows S4D system has the potential for wide applications and tested S4D system for different protein targets at various cell compartments. Moreover, the manuscript tested application of S4D system in TNF- α / NF- κ B dependent cell death process using a knock in (KI) approach for I κ B α or RelA protein degradation.

Overall, experiments are well designed and properly controlled. Moreover, S4D system could be used to study wide variety of degradation pathways using IMiDs, akin to auxin system. S4D system can also be exploited to analyze degradation of endogenously tagged proteins in genome editing experiments. This platform exploits the advantage of endogenous CRBN ligase, and therefore could mitigate the use of many non-physiological variables in experiments under investigation using such approaches.

Specific Points:

1. Is AGIA-S4D-WT degradation responsive to CRBN expression levels at 10 μ M IMiDs (Fig. 1d)?
2. Is endogenous CRBN sensitive to MLN treatment like exogenously expressed (Figs 1 and 2)? Is exogenous CRBN also sensitive to IMiDs (Fig. 1c, d)?
3. Figure 2b could be more meaningful +/- IMiDs. However, immunoblots are shown S4b? Is DRD1 entirely plasma membrane localized?
4. Explain the differential mobilization of N- or C-terminally tagged p53 with S4D (Fig. 2e).
5. Is there significant difference in cGAMP mediated increase in the luciferase activity seen in STING-AGIA vs STING-S4D-AGIA (Fig. 3a, bars 5vs8)?
6. Some references appear the end of manuscript. Fix it.

Reviewer #3 (Remarks to the Author):

Recommendation: Publish after minor revisions

Comments:

In this paper, the authors described the development of new protocols for achieving protein knockdown with chemicals. By using Immunomodulatory drugs (IMiDs), selective protein degradation can be attained through chemical induced interacting with celebron (CRBN) in the cullin E3 ubiquitin ligase complex (CRL4CRBN). The authors discovered the IMiD-dependent Sal-like protein 4 (SALL4) degron (S4D) system for chemical protein knockdown. In RelA-S4D KI cells, authors showed S4D system can be employed as a tool for elucidating potential TNF- α and RelA-involved molecular mechanism. Generally, I feel this paper presents a nice work about small molecule based protein degradation for understanding the functions of the POI. The topic is important and I feel the work is well organized and written by the authors. Although there is a similar work published last year in PNAS, I think more practical tools for biology study should be welcomed in this newly emerging field. The paper can be considered for publication after some minor revisions:

1) The authors should mention and discuss PROTAC, which will give a broader and better depiction of whole field for readers. Although PROTAC is not as easily be used as this type of knockdown system if a suitable POI binder is missing, PROTAC has a great advantage over other systems as it can work without incorporation of fusion protein such as tag or recognition binding domain. Three following reviews about protein degradation and PROTAC should be cited in the case.

a) Multispecific drugs herald a new era of biopharmaceutical innovation. *Nature*, 2020, 580:329-338. <https://doi.org/10.1038/s41586-020-2168-1>

b) PROTACs: great opportunities for academia and industry, *Signal Transduction and Targeted Therapy*, 2019, 4, 64-96, DOI:10.1038/s41392-019-0101-6.

c) Targeted protein degradation: expanding the toolbox, *Nature Reviews Drug Discovery*. 2019 18(12):949-963

2) If Golgi-localized protein knockdown can be achieved by this new method ?

3) Authors should compare the advantage and efficiency of their method with other approaches including PROTAC in more details.

Point-by-Point Responses to the Reviewers' Critiques (COMMSBIO-20-0688-T)

We deeply appreciate the thorough analysis and constructive suggestions provided by the three reviewers to guide us to further improve our manuscript. As described in more detail below, we have addressed all the reviewers' concerns. With this extensive revision, we hope that the reviewers will concur with us that we have addressed all of the raised concerns in a satisfactory manner and, consequently, substantially strengthened our paper.

Reviewer #1

The authors introduce another method for targeted protein degradation using a SAL4 ZnF degron protein tag fusion and IMiDs. This closely resembles previous work using an IKZF3 tag, but will be of new interest due to its success in N-terminal fusions and a variety of substrates with diverse subcellular localisation. Specific comments are given below.

Response: We thank the reviewer for the kind comments on new interest of our manuscript.

Results text

Line 73. The authors should state that they are co-transfecting CRBN to allow degradation in these cells otherwise the results as stated don't make sense (I don't see why the readers should hunt through the figure legend to find basic points on the experimental hypotheses).

Response: We thank the reviewer for the kind suggestion about explanation on several experiments in our manuscript. We described in the revised manuscript that we used CRBN^{-/-} HEK293T cells expressing FLAG-CRBN to induce protein degradation of Venus-S4D (lines 77-79).

Line 80. Typo MNL4924

Response: We apologize for this mistake in the original manuscript. We fixed in the revised manuscript (line 82).

Line 133. "...suggesting that the degradation of DRD1 occurs on the plasma membrane". If internalised the DRD1 N-terminus would still be inside endosomes

or lysosomes that may be inaccessible to CRBN. Thus, I am not sure the statement that DRD1 degradation must happen at the plasma membrane is strictly true.

Response: We thank you for the outstanding comments about protein degradation of DRD1. We fully agree the reviewer's comment that it is uncertain whether protein degradation of DRD1 occur at plasma membrane. To strengthen our suggestion that the S4D system can be applied for protein localized at plasma membrane, we produced HeLa cells expressing DRD1-S4D-AGIA stably and investigated localization and degradation of DRD1-S4D-AGIA in the stable HeLa cells. As results, DRD1-S4D-AGIA was almost localized at plasma membrane (Supplementary Fig. 4a in the revised manuscript) and IMiD treatment induced protein degradation of DRD1-S4DAGIA (Supplementary Fig. 4b in the revised manuscript). We thought that these results strongly suggest that S4D-tagged proteins localized at plasma membrane can be degraded by IMiD treatment. However, as the reviewer's comments, these results do not completely sure that the degradation occur at plasma membrane if the degradation occur inside endosomes or lysosomes. Therefore, we weaken and modified our representation about DRD1 degradation in the revised manuscript (lines 151-157).

Line 159. "The S4D system is therefore a useful tool for analyzing the cellular functions of target proteins in transient expression experiments". In practise, the authors example is greatly aided by the having cells that lack endogenous expression of the target protein.

Response: We thank the reviewer for pointing out important point to apply the S4D system for transient assay. We completely agree with this opinion that the S4D system is useful tool in the case of deficiency of target protein in the cells. As reviewer's suggestion, we added the condition to apply the S4D system in discussion section in the revised manuscript (lines 343-347).

Figures

The font size used for text within the figures is far far too small throughout (it is a struggle to read – e.g. especially fig 5d label "pretreated pomalidomide" needed 200% zoom to read).

Response: We apologize for the difficulty to read the text in figure. As according to the reviewer's comment, we modified all figures to make it easier to read in the revised manuscript.

Fig 1C legend. Abbreviation "DM" should be noted as used for DMSO.

Response: We apologize for the lack of explanation about abbreviation in figures. As according to the reviewer's comment, we explained all abbreviation in figure legends.

Why is CRBN band poorly seen in DMSO or thalidomide-treated cells? It is not ideal to judge neosubstrate levels when the control levels of E3 are so varied.

Response: We thank the reviewer for bringing up the important issue on the expression level of overexpressed CRBN. Thalidomide and its derivatives (IMiD) increase the expression of overexpressed CRBN (Fig. 1c-e in the revised manuscript), and the variation of overexpressed CRBN was also observed in the previously study (*Krönke, J. et al. 2014. Science. 343:301–305*). In previous research, IMiD affects on polyubiquitination level of CRBN but it remains unclear whether the autoubiquitination on CRBN is a reason on variation of overexpressed CRBN (*Ito, T. et al. 2010. Science. 327:1435–1350*). Most importantly, the variation of expression was not observed in the case of endogenous CRBN in both our results and previous reports (*Krönke, J. et al. 2014. Science. 343:301–305; Krönke, J. et al. 2015. Nature. 523:183–188*).

To show that degradation of S4D-tagged protein is not affected by variation of overexpressed CRBN, we performed additional experiments using parental HEK293T cells and the results added in the revised manuscript. As a result, IMiD treatment induced protein degradation of Venus-S4D without affecting expression of endogenous CRBN (Fig. 1f in the revised manuscript). In addition, Venus-S4D-G416A mutant was not induced protein degradation by IMiD treatment although expression of overexpressed CRBN was similarly affected by IMiD treatment (Fig. 1e in the revised manuscripts). These results strongly suggest that protein degradation of S4D-tagged protein is scarcely influenced by variation of overexpressed CRBN.

Fig2D. Why does POM stabilise MAVS in CRBN-/- cells? Is POM binding to the SAL4 tag helping to give a thermostabilisation effect to the fusion protein? Is this a

sign of side effects from the S4D system that future users should be made aware of?

Response: We thank the reviewer for pointing out the concern on side effects derived from S4D system. To confirm whether stability of MAVS is the artificial effect from S4D system, we performed the same experiments as Fig. 2d (original manuscript) repeatedly. As results, pomalidomide did not stabilize the expression level of MAVS in all experiments. Therefore, we concluded that the stabilization of MAVS in original manuscript was an experimental error and added the new result in the revised manuscript (Fig. 2e in the revised manuscript).

Methods

Plasmids section. Small differences in construct design appear to effect the ability of the S4D system to work as judged in Fig 1c by the failure of the m3 degron motif. The authors do not describe the restriction sites employed or if spacer amino acids are introduced (e.g. small Gly, Ala or Ser type a.a.) between tag and fusion target protein. I can imagine these elements might effect the outcome. As such, it is impossible for any reader to reproduce the constructs described in this paper. Thus, further details are required. Best would be a supplemental figure showing the cloning map. Otherwise the methods appear quite detailed.

Response: We apologize for these deficiencies in the original manuscript. We fully agree the reviewer's comment that short spacer sequence may affect protein degradation of the target protein. In the regard with spacer amino acids sequence between S4D tag and target protein, it was reported that I3D-dependent protein degradation was not affected by length of the spacer sequence (*Koduri, V. et al. 2019. Proc. Natl. Acad. Sci. U.S.A. 116:2539-2544*). Because the S4D tag compose of single zinc finger domain same as I3D tag, we thought that short spacer sequences between S4D tag and target protein do not affect protein degradation. Therefore, we used cloning vectors having a several spacer sequence for restriction enzymes. Actually, all S4D-tagged proteins used in this study were induced protein degradation by IMiD treatment, indicating that short spacer scarcely affect protein degradation on the S4D system. As according to reviewer's comment, we showed the spacer amino-acid sequences in all figures in the revised manuscript.

Discussion.

The authors mention one advantage of their system is the efficacy with thalidomide versus the IKZF3 tag. However, the S4D tag results suggest that 10 micromolar pomalidomide is the only treatment that gives “near” complete knockdown, whereas thalidomide is the least effective. I imagine the authors would like their system to be widely adopted, used for other publications and cited. Therefore, the authors should comment on their recommendations to future users of the conditions they recommend (choice of IMiD, treatment time etc). They should comment on the specificity issues of using 10 micromolar pomalidomide.

Response: We thank you for your outstanding suggestion about neosubstrate specificity on the S4D system. We fully agree the reviewer’s comment that pomalidomide is more effective IMiD for the S4D system than thalidomide. In addition, we also agree the reviewer’s comment that we should mention the usefulness of S4D system. In the revised manuscript, we performed additional experiments using 5-hydroxythalidomide (5-HT), which is a thalidomide metabolite. In previous study, we showed that 5-HT strongly induces protein degradation of SALL4 and IKZF1 is not induced protein degradation. As shown Fig. 7c-e in the revised manuscript, both efficiency of SALL4 degradation and specificity on IKZF1 is conserved in the S4D system. More importantly, degradation potency of 5-HT on S4D-tagged protein was the same level as that of pomalidomide (Fig. 7f in the revised manuscript). Because it has not been reported that 5-HT has more diverse neosubstrates than thalidomide, we believe that the combination 5-HT and the S4D system is a more useful tool. Based on these new results, we added new Figure 7 and discussed kind of IMiD to use and specificity in the revised manuscript (lines 311-331 and 389-409).

Reviewer #2

IMiDs such as thalidomide and its derivatives have been shown to regulate CRBN-ubiquitin ligase dependent (neo)substrate degradation, e.g., IKZF3 and Sal-like protein 4 (SALL4). This manuscript reports development of a Sal-like protein 4 (SALL4) degron (S4D) system, a binding interface for CRBN ubiquitin ligase in presence of IMiDs, for the use of protein substrate degradation via Cullin/UPS. The manuscript shows S4D system has the potential for wide applications and tested S4D system for different protein targets at various cell compartments. Moreover, the manuscript tested application of S4D system in TNF- α / NF- κ B dependent cell death process using a knock in (KI) approach for I κ B α or RelA protein degradation.

Overall, experiments are well designed and properly controlled. Moreover, S4D system could be used to study wide variety of degradation pathways using IMiDs, akin to auxin system. S4D system can also be exploited to analyze degradation of endogenously tagged proteins in genome editing experiments. This platform exploits the advantage of endogenous CRBN ligase, and therefore could mitigate the use of many non-physiological variables in experiments under investigation using such approaches.

Response: We thank you for kind comments on the advantage of S4D system.

Specific Points:

1. Is AGIA-S4D-WT degradation responsive to CRBN expression levels at 10 μ M IMiDs (Fig. 1d)?

Response 1: We thank the reviewer for bring up the important concern on CRBN expression levels for protein degradation of S4D-tagged protein. Consistent with previous report (*Krönke, J. et al. 2014. Science. 343:301–305*), expression levels of overexpressed CRBN was increased by IMiD treatment also in our study (Fig. 1c-e). However, previous reports (*Krönke, J. et al. 2014. Science. 343:301–305; Krönke, J. et al. 2015. Nature. 523:183–188*) and our results showed that expression level of endogenous CRBN was not affected by IMiD treatment and S4D-tagged proteins were induced IMiD-dependent protein degradation (Fig. 4d-f in the revised manuscript). To show that protein degradation of S4D-tagged proteins is not responsive to CRBN expression levels, we additionally performed experiment using parental HEK293T cells

and added the result Fig. 1f in the revised manuscript. As a result, expression levels of endogenous CRBN was not increased by IMiD treatment and AGIA-Venus-S4D was degraded (Fig. 1f in the revised manuscript). Furthermore, as shown Fig. 1e in the revised manuscript. Venus-S4D-G416A mutant was not induced protein degradation by IMiD treatment although expression of overexpressed CRBN was similarly affected by IMiD treatment (Fig. 1e in the revised manuscripts). These results strongly suggest that protein degradation of S4D-tagged protein is scarcely influenced by expression levels of overexpressed CRBN. As according to this comment, we added new experiment result (Fig. 1f) and mentioned (lines 88-93) in the revised manuscript.

2. Is endogenous CRBN sensitive to MLN treatment like exogenously expressed (Figs 1 and 2)? Is exogenous CRBN also sensitive to IMiDs (Fig. 1c, d)?

Response 2: We thank the reviewer for suggestion on endogenous CRBN expression levels IMiD or MLN4924 treatment. To address the reviewer's comments, we performed same experiments as Fig. 4d and investigated endogenous CRBN expression levels. As shown Fig. 4d in the revised manuscript, endogenous CRBN expression levels were scarcely affected by MLN4924 treatment. In regards with exogenous CRBN sensitivity to IMiDs, it was reported that polyubiquitination level of endogenous CRBN was affected by interaction with IMiDs (*Ito, T. et al. 2010. Science. 327:1435–1350*). However, it is still unclear whether this effect on polyubiquitination on CRBN is reason to increase exogenous CRBN expression levels.

Because it was showed that protein degradation of S4D-tagged proteins was scarcely influenced by variation of exogenous CRBN as described above (*Response 1*), we believe that effect on exogenous CRBN expression levels do not alter the results of S4D-tagged protein degradation.

3. Figure 2b could be more meaningful +/- IMiDs. However, immunoblots are shown S4b? Is DRD1 entirely plasma membrane localized?

Response 3: We thank you for raising important suggestion about protein degradation at various cellular compartments. We fully agree the reviewer's comments that immunostaining experiments is more meaningful in the presence of IMiD. However, the S4D-tagged proteins were detected as transiently exogenous protein the in Fig. 2b. Therefore, it was experimentally difficult to trace the protein degradation of the S4D-tagged proteins because the protein synthesis of S4D-tagged proteins goes on

occurring in the cells strongly. In conclusion, we argue that the overexpressed proteins were mainly localized at proper cell compartments, and immunoblot analysis showed that IMiD-dependent degradations of the S4D-tagged proteins were induced.

As according to reviewer's comment, we moved the result of Supplementary Fig. 4b in original manuscript to Fig. 2c in the revised manuscript.

We also recognize that overexpressed DRD-S4D-AGIA is not fully localized at plasma membrane. Therefore, we weakened our representation that protein degradation of DRD1 was caused at plasma membrane (lines 133 and 154-157). To strengthen our suggestion that S4D system can be applied for target protein localized at various cellular compartments, we produced HeLa cells expressing DRD1-S4D-AGIA stably and investigated localization and degradation of DRD1-S4D-AGIA in the stable HeLa cells. As results, DRD1-S4D-AGIA is almost localized at plasma membrane (Supplementary Fig. 4a) and IMiD treatment induced protein degradation of DRD1-S4D-AGIA (Supplementary Fig. 4b), supporting that the S4D system can degrade DRD1. As according to this comment, we added new experiment results (Supplementary Fig. 4a and b) and mentioned (lines 151-157) in the revised manuscript.

4. Explain the differential mobilization of N- or C-terminally tagged p53 with S4D (Fig. 2e).

Response 4: We thank for the comment about important issue on difference of N- or C-terminally S4D-tagged proteins. We also recognize the differential mobilization between S4D-protein and protein-S4D (Fig. 2j in the revised manuscript). Spacer sequences between S4D tag and each protein is different between N- and C-terminally tagged protein as shown in each figure. However, in Fig. 2j (p53), Fig. 2k (DRD1), or Supplementary Fig. 2d (Venus), the differential mobilization is not same among these S4D-tagged proteins. Therefore, we think that the differences do not result from artificial effect by construction of expression vector but we cannot fully explain the reason.

5. Is there significant difference in cGAMP mediated increase in the luciferase activity seen in STING-AGIA vs STING-S4D-AGIA (Fig. 3a, bars 5vs8)?

Response 5: We thank the reviewer for comments about transient experiment on STING. As according to the reviewer's comment, we investigated significant difference between STING-AGIA and STING-S4D-AGIA (Fig. 3a in the revised manuscript). As

a result, cGAMP-dependent activation of STING-S4D-AGIA was stronger than that of STING-AGIA. However, given that this experiment is a transient assay we believe that transient assay using S4D system is qualitative assay rather than quantitative assay. Therefore, it is the most important result that pomalidomide treatment suppressed activation of STING-S4D-AGIA (bar 8 and 9 in Fig. 3a) but did not that of STING-AGIA (bar 5 and 6 in Fig. 3a).

6. Some references appear the end of manuscript. Fix it.

Response 6: We apologize for errors on manuscript format. We fixed the references positions in the revised manuscript.

Reviewer #3

Recommendation: Publish after minor revisions

Comments:

In this paper, the authors described the development of new protocols for achieving protein knockdown with chemicals. By using Immunomodulatory drugs (IMiDs), selective protein degradation can be attained through chemical induced interacting with celebren (CRBN) in the cullin E3 ubiquitin ligase complex (CRL4CRBN). The authors discovered the IMiD-dependent Sal-like protein 4 (SALL4) degron (S4D) system for chemical protein knockdown. In RelA-S4D KI cells, authors showed S4D system can be employed as a tool for elucidating potential TNF- α and RelA-involved molecular mechanism. Generally, I feel this paper presents a nice work about small molecule based protein degradation for understanding the functions of the POI. The topic is important and I feel the work is well organized and written by the authors. Although there is a similar work published last year in PNAS, I think more practical tools for biology study should be welcomed in this newly emerging field. The paper can be considered for publication after some minor revisions:

Response: We thank the reviewer for kind comments about the S4D system on chemical protein knockdown.

1) The authors should mention and discuss PROTAC, which will give a broader and better depiction of whole field for readers. Although PROTAC is not as easily be used as this type of knockdown system if a suitable POI binder is missing, PROTAC has a great advantage over other systems as it can work without incorporation of fusion protein such as tag or recognition binding domain. Three following reviews about protein degradation and PROTAC should be cited in the case.

a) Multispecific drugs herald a new era of biopharmaceutical innovation. Nature, 2020, 580:329-338. <https://doi.org/10.1038/s41586-020-2168-1>

b) PROTACs: great opportunities for academia and industry, Signal Transduction and Targeted Therapy, 2019, 4, 64-96, DOI:10.1038/s41392-019-0101-6.

c) Targeted protein degradation: expanding the toolbox, Nature Reviews Drug Discovery. 2019 18(12):949-963

Response: We thank the reviewer for outstanding suggestion to improve our manuscript. We fully agree the reviewer's opinion that PROTAC is a very powerful approach because PROTAC does not require for additional sequence on target protein. As according to the reviewer's suggestion, we added a new paragraph in discussion section describing each advantage and disadvantage with comparing PROTAC, the S4D system, and other systems developed previously (lines 370-388).

2) If Golgi-localized protein knockdown can be achieved by this new method ?

Response: We thank the reviewer for the kind suggestion about protein degradation of S4D-tagged proteins at various cellular compartments. As according to reviewer's comment, we evaluated protein degradation of GM130, which is known as a protein localized at Cis-Golgi Network (CGN) and added several results in the revised manuscript (Fig. 2f-i). Immunoblot analysis showed that IMiD-dependent protein degradation of GM130-S4D-AGIA was observed (Fig. 2h and i). In addition, localization of overexpressed GM130-S4D-AGIA was confirmed by immunofluorescence staining using GM130 specific antibody (Fig. 2g). These results strongly suggest that S4D system can be applied for target protein localized at Golgi apparatus. As according to this comment, we added new experiment results (Fig. 2g-i) and mentioned (lines 139-147) in the revised manuscript.

3) Authors should compare the advantage and efficiency of their method with other approaches including PROTAC in more details.

Response: We thank the reviewer for kind suggestion to improve our manuscript. We believe that the S4D system is a powerful approach because of the short additional sequence and the use of IMiD which is most characterized molecular glue, if there is not chemical compound available for PROTAC. To strengthen usefulness of the S4D system, we performed additional experiments using 5-Hydroxythalidomide (5-HT). 5-HT can induced strong protein degradation of S4D-tagged (Fig. 7c-f in the revised manuscript) and cannot induced protein degradation of I3D-tagged protein and endogenous IKZF1/3 (Fig. 7f in the revised manuscript) (*Yamanaka, S. et al. 2020. bioRxiv. doi: 10.1101/2020.02.28.969071*). Based on these results, we added new Figure 7 and described advantages of the S4D system and other systems in discussion section (lines 311-331 and 389-409).

REVIEWERS' COMMENTS:

Reviewer #1 (Remarks to the Author):

The authors have addressed my comments. There is a typo at line 969 "RelA-sfGFP-S43D" (should delete the "3").

Reviewer #2 (Remarks to the Author):

Authors have performed new experiments, and addressed all questions satisfactorily. However, the response to point#4, about differential mobilization of N- or C-terminally tagged p53 with S4D is somewhat weak. Authors may mention in the text for the specific experiments that N-or-C-terminally tagged p53 with S4D mobility is different for some unknown reasons.

I have no questions further.

Reviewer #3 (Remarks to the Author):

The authors have addressed questions and revised their paper accordingly. I would like to suggest it for publication.

Point-by-Point Responses to the Reviewers' Critiques (COMMSBIO-20-0688-T)

We deeply appreciate the constructive suggestions and kind comment to publish this study. As described in more detail below, we have addressed all the reviewers' concerns.

Reviewer #1

The authors have addressed my comments. There is a typo at line 969 "RelA-sfGFP-S43D" (should delete the "3").

Response: We thank the reviewer for the kind comments on our revised manuscript. As according to the reviewer's comment, we modified the typo at line 1010 in the revised manuscript.

Reviewer #2 (Remarks to the Author):

Authors have performed new experiments, and addressed all questions satisfactorily. However, the response to point#4, about differential mobilization of N- or C-terminally tagged p53 with S4D is somewhat weak. Authors may mention in the text for the specific experiments that N-or-C-terminally tagged p53 with S4D mobility is different for some unknown reasons.

I have no questions further.

Response: We thank the reviewer for the kind suggestion about differential mobilization of N- or C-terminally tagged proteins. As according to the reviewer's comment, we mentioned differential mobilization of N- or C-terminally tagged proteins in the revised manuscript (lines 161-162).

Reviewer #3 (Remarks to the Author):

The authors have addressed questions and revised their paper accordingly. I would like to suggest it for publication.

Response: We thank the reviewer for the kind comments on our revised manuscript.